

# Estimation of mechanistic parameters in the gas-phase reactions of ozone with alkenes for use in automated mechanism construction

Mike J. Newland[1,a], Camille Mouchel-Vallon[1,b], Richard Valorso[2], Bernard Aumont[2], Luc Vereecken[3], Michael E. Jenkin[4], Andrew R. Rickard[1,5]

[1] Wolfson Atmospheric Chemistry Laboratories, Department of Chemistry, University of York, United Kingdom
[2] Univ Paris Est Creteil and Université de Paris, CNRS, LISA, F-94010 Créteil, France.
[3] Forschungszentrum Jülich GmbH, Institute for Energy and Climate, IEK-8 Troposphere, 52428 Jülich, Germany
[4] Atmospheric Chemistry Services, Okehampton, Devon, EX20 4QB, United Kingdom.
[5] National Centre for Atmospheric Science, Wolfson Atmospheric Chemistry Laboratories, University of York, United Kingdom.

[a] now at: ICARE-CNRS, 1 C Av. de la Recherche Scientifique, 45071 Orléans Cedex 2, France.
[b] now at: Laboratoire d'Aérologie, Université de Toulouse, CNRS, UPS, Toulouse, France.

*Correspondence to*: Mike Newland (mike.newland@gmail.com)  and Andrew Rickard (andrew.rickard@york.ac.uk).

**Abstract.** Reaction with ozone is an important atmospheric removal process for alkenes. The ozonolysis reaction produces carbonyls, and carbonyl oxides (Criegee intermediates, CI), which can rapidly decompose to yield a range of closed shell and radical products, including OH radicals. Consequently, it is essential to accurately represent the complex chemistry of Criegee intermediates in atmospheric models in order to fully understand the impact of alkene ozonolysis on atmospheric composition. A mechanism construction protocol is presented which is suitable for use in automatic mechanism generation. The protocol defines the critical parameters for describing the chemistry following the initial reaction, namely: the primary carbonyl / CI yields from the primary ozonide fragmentation; the amount of stabilisation of the excited CI (CI*); the unimolecular decomposition pathways, rates and products of the CI; the bimolecular rates and products of atmospherically important reactions of the stabilised CI (SCI). This analysis implicitly predicts the yield of OH from the alkene-ozone reaction. A comprehensive database of experimental OH, SCI and carbonyl yields has been collated using reported values in the literature and used to assess the reliability of the protocol. The protocol provides estimates OH, SCI and carbonyl yields with a root mean square error of 0.13 and 0.12 and 0.14, respectively. Areas where new experimental and theoretical data would improve the protocol and its assessment are identified and discussed.

## 1    Introduction

Reaction with ozone is an important atmospheric removal process for alkenes, competing with reaction with OH and $NO_3$ radicals. The ozonolysis reaction produces carbonyls and carbonyl oxides, commonly denoted Criegee intermediates (CI), which can rapidly rearrange or decompose to yield a range of closed shell and radical products (Johnson and Marston, 2008). Alkene ozonolysis has been shown to be an important non-photolytic source of OH radicals, with field measurements (Paulson and Orlando, 1996; Elshorbany et al., 2009) and modelling studies (e.g. Bey et al., 1997) suggesting it to be the dominant tropospheric OH source at night, in the winter (Heard et al., 2004; Emmerson et al., 2005), and in indoor environments (Carslaw, 2007). Unimolecular CI reactions (Ehn et al. 2014; Iyer et al., 2020) and bimolecular reactions of Stabilised Criegee Intermediates (SCI), with e.g. organic acids and peroxy radicals (e.g. Kristensen et al., 2014; Sakamoto et al., 2013; Zhao et al., 2015; Mackenzie-Rae

et al., 2016), have been implicated in secondary organic aerosol formation. SCI can also act as an oxidant, this
has been studied particularly for the reaction with $SO_2$ (e.g. Welz et al., 2012, Mauldin II et al., 2012; Caravan et
al., 2020) which can lead to sulfate aerosol production and hence impact radiative forcing and climate (Pierce et
al. 2013; Percival et al. 2013). However, both the $SO_2$ and organic acid reactions, while important locally, are
likely only of minor importance to global budgets of sulfate aerosol and organic acids (Welz et al., 2014; Newland
et al., 2018). The dominant removal processes for most SCI in the troposphere are reaction with water vapour or
unimolecular reaction (Vereecken et al., 2017). However, for certain structures, these reactions are sufficiently
slow for bimolecular reactions with other trace gases to become important.

Understanding of the complex nature of the chemistry of Criegee intermediates has progressed rapidly

in recent years, particularly with regard to the mechanisms and rates of decomposition of CI*/SCI, and the
bimolecular reaction rates of SCI. This has been facilitated by: direct experimental measurements of CI kinetics,
generating CI through photolysis of di-iodo precursors (e.g. Welz et al., 2012; Chhantyal-Pun et al. 2020, and
references therein); indirect measurements of CI kinetics during alkene ozonolysis experiments (e.g. Berndt et al.
2014a, 2014b, 2015; Newland et al., 2015), and extensive theoretical studies (e.g. Vereecken et al., 2017, and
references therein).

The reaction of ozone with alkenes proceeds by a concerted addition to the C=C double bond, forming a

short lived Primary Ozonide (POZ). Typically, the POZ fragments into two pairs of carbonyls and Criegee
intermediates (CI) (Figure 1); for small to medium sized alkenes ($C_{\leq 10}$) this POZ is vibrationally excited,
decomposing promptly, while for large alkenes (e.g. $C_{\geq 15}$, sesquiterpenes), theoretical studies suggest that the POZ
can be collisionally stabilized prior to decomposition (Chuong et al., 2004; Nguyen et al., 2009a). Theoretical
work also indicates that a small fraction of the POZ can rearrange to a carbonyl-hydroperoxide when vinylic H-
atoms are present (Pfeifle et al., 2018); this mechanism is discussed separately below. It has also been suggested
that different pathways may play a more significant role for a small number of systems e.g. cyclohexadienes
(Pinelo et al., 2013).




**Figure 1. First step of alkene ozonolysis. A primary ozonide (POZ) is formed which rapidly decomposes to yield a pair**
**of chemically activated Criegee intermediates and carbonyl products.**

Criegee intermediates are generally zwitterionic in nature, as shown in Figure 1, but the moiety is denoted

simply as a >COO structure below (not to be confused with alkylperoxy radicals, ROO$^{\bullet}$). CI can be formed with
the terminal oxygen of the carbonyl oxide moiety in either an *E* (*anti*) or *Z* (*syn*) configuration relative to a given
substituent group. The two conformers are not in rapid equilibrium, with quantum calculations showing that the
energy barrier to rotational interconversion for $CH_3CHOO$ is about 29 kcal mol$^{-1}$ (Johnson and Marston, 2008,
and references therein); this was confirmed by Vereecken et al. (2017) who calculated barriers exceeding 30 kcal



mol$^{-1}$ for saturated CI conformers. Isomeric CI conformers have been shown to have different unimolecular
reaction rates (e.g. Vereecken et al., 2017), follow different unimolecular pathways (Herron and Huie, 1977; Niki
et al., 1987; Martinez and Herron, 1987; Kidwell et al., 2016), and have very different reaction rates with water
(e.g. Taatjes et al., 2013; Sheps et al., 2014; Huang et al., 2015). Therefore, these conformers must necessarily be
considered as separate species, irreversibly partitioned according to their nascent ratios, to accurately represent
the effects of alkene ozonolysis on atmospheric composition.

Structure Activity Relationships (SARs) are commonly used to design the protocols needed to develop

automated mechanism generation tools (Vereecken et al., 2018). This paper forms part of a series of articles
devoted to the development of SARs for mechanism generation (Jenkin et al., 2018a, 2018b, 2019, 2020). Updated
SAR methods for the initial reactions of O$_3$ with unsaturated organic compounds are presented in a companion
paper (Jenkin et al., 2020), while in this work, a protocol is presented for the subsequent chemistry occurring
following the initial O$_3$ addition. This details the yields of carbonyls and Criegee intermediates from the alkene +
O$_3$ reaction, and the subsequent fate of the Criegee intermediates, and accounts for the minor pathway by carbonyl-
hydroperoxide radical formation. The protocol is based on available experimental data and theoretical data
combined. For areas in which limited data exists, the protocol is set up to be easily updated as new experimental
or theoretical results become available. These areas are highlighted in the paper and are recommended areas of
further research. The protocol is currently being used to guide development of alkene ozonolysis chemistry in the
Generator for Explicit Chemistry and Kinetics of Organics in the Atmosphere, GECKO-A (Aumont et al., 2005),
and the Master Chemical Mechanism, MCM (Jenkin et al., 1997, Saunders et al., 2003). It is noted that the protocol
does not currently consider aromatic species that have been shown to react with ozone, such as catechols, for
which the mechanism may be different to the Criegee mechanism described here.

The methodology for applying the protocol described in this work is summarised in Figure 2. The initial

addition of ozone to the double bond follows the protocol described in the companion paper (Jenkin et al., 2020).
The POZ formed from this protocol then decomposes according to the rules determined in Section 2, to give the
primary carbonyl and the CI yields ($\alpha$), and possibly a minor fraction of carbonyl-hydroperoxide. A fraction ($\gamma$)
of the CI is then stabilised (Section 3). Both the stabilised and chemically activated CI then follow the relevant
set of rules from Vereecken et al. (2017) to ascribe them unimolecular decomposition mechanisms (and hence
products) and rates (Section 4), and bimolecular reaction rates with water vapour (Section 5). Finally, bimolecular
reaction rates with other atmospherically important species are assigned as a function of the SCI structure (Section

5).






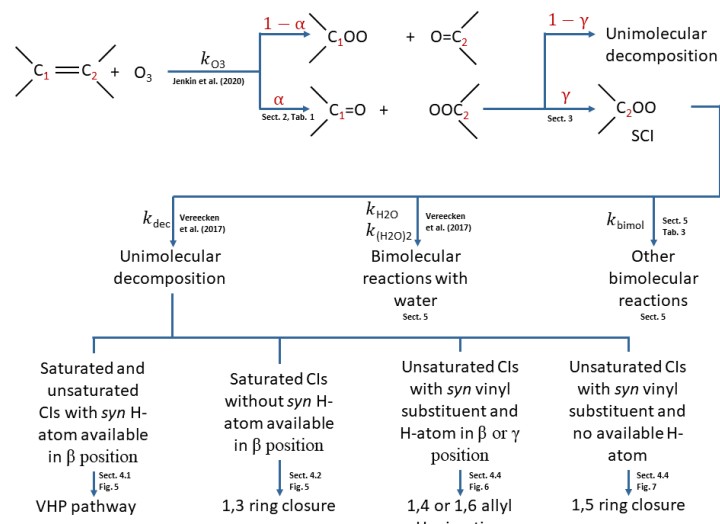


**Figure 2. Flow diagram for implementation of the protocol. α = branching ratios in POZ decomposition, γ = fraction of CI stabilised. >COO denotes the Criegee intermediate formed.**

## 2 Primary Ozonide Fragmentation

### 2.1 Alkenes with aliphatic substituents

The fragmentation of the POZ has previously been parameterized based on the branching pattern around the double bond of the parent alkene (Jenkin et al., 1997; Rickard et al., 1999). Generally, it can be said that there is a preference for formation of the more substituted CI, e.g. the ozonolysis of 2-methyl propene yields ~0.7 $(CH_3)_2COO$ and ~0.3 $CH_2OO$ (Rickard et al., 1999). However, consideration of just the immediate substituents of the double bond breaks down for more complex structures and for oxygenated substituents. There is clearly also an effect of substitution around the carbon adjacent to the double bond. For instance, when there is a *t*-butyl group attached to the double bond, a strong preference is seen for formation of the opposing CI, as observed for yields of trimethylacetaldehyde from 3,3-dimethyl-1-butene (0.67) and trans-2,2-dimethyl-3-hexene (0.84) (Grosjean and Grosjean, 1997). Using data from Grosjean and Grosjean (1997), various homologous series of alkenes can be considered, such as the series with increasing methyl substitution on the α-carbon. For the 1-alkene series (Figure 3), yields of the larger carbonyl of 0.35, 0.51, and 0.67 are determined for 1-butene, 3-methyl-1-butene, 3,3-dimethyl-1-butene respectively.



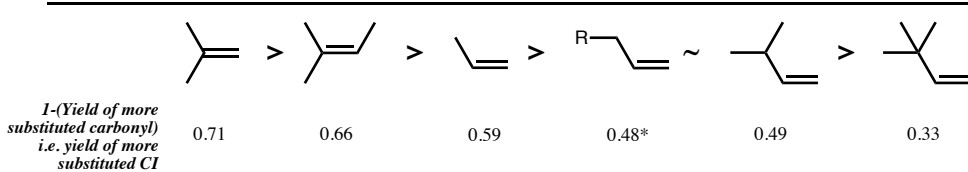

| | | | | | | |
|---|---|---|---|---|---|---|
| **1-(Yield of more substituted carbonyl) i.e. yield of more substituted CI** | 0.71 | 0.66 | 0.59 | 0.48* | 0.49 | 0.33 |

**Figure 3. Decreasing order of preference, from left to right, of more substituted CI formation from ozonolysis of example alkyl substituted alkenes. Values are 1 – (mean of measured yields of carbonyls) (Spreadsheet S1). * Mean measured yield of propanal (i.e. 1 – more substituted CI) formation from 1-butene is 0.64, but all other 1-alkenes are 0.45 – 0.50.**

Such relationships have been observed and discussed previously by Grosjean and Grosjean (1997) in terms of: (i) steric hindrance potentially weakening the O-O bond in the POZ on the side of the bulky substituent, and (ii) the inductive effect of adjacent alkyl groups strengthening the O-O bonds in the POZ (Grosjean and Grosjean, 1997). Earlier work considering POZ fragmentation in the aqueous phase (Fliszár and Renard, 1970; Fliszár and Granger, 1970; Fliszár et al., 1971) described similar relationships to those observed in the gas phase (i.e. that shown in Figure 3), except in the case of terminal alkenes, for which the reverse trend was observed. In these studies, the observed trends are discussed in terms of stabilisation of the positive charge on the carbon in the POZ through: (i) 'hyperconjugative stabilisation' in the transition state, and (ii) the inductive effect during the POZ cleavage, with steric effects discounted as being unimportant in determining the POZ fragmentation pattern.

These works can be summarised by saying that it appears that a substituent with a partial negative charge, such as a methyl group, can stabilise the positive charge on the adjacent carbon in the POZ. This leads to a greater yield of the CI containing the more stabilising substituents. On the other hand, a substituent that leads to a partial positive charge on the $\alpha$-carbon leads to a lower yield of that CI.

## 2.2 Oxygenated alkenes

Following the rationale discussed above, oxygenated substituents on the $\alpha$-carbon might be expected to strongly influence the primary ozonide fragmentation pattern. The number of product yield studies on the ozonolysis of most classes of unsaturated oxygenates is rather limited. As discussed below, some oxygenated substituents appear to destabilise the positive charge on the carbon in the POZ (i.e. disadvantaging POZ fragmentation towards the production of the CI on the oxygenated side), particularly carbonyl groups, while others such as acrylate esters and carboxylic acids may stabilise the CI, favouring its formation. However, data is very limited and often ambiguous for most of the oxygenated classes. This is partly due to challenges in measuring products containing multiple oxygenated groups, partly that some of these classes are likely to be present in negligible amounts in the atmosphere and, for some, that ozonolysis will be a negligible atmospheric sink compared to e.g. reaction with OH or photolysis. The available data is provided in the Supplement, Spreadsheet S1.

### 2.2.1 Enones / enals

Primary carbonyl yields have been reported for two $\alpha$-$\beta$ terminally unsaturated ketones ($H_2C=CHC(O)R$). For methyl vinyl ketone (MVK), Grosjean et al. (1993) and Ren et al. (2017) determined a strong preference for formation of the ketone substituted product methyl glyoxal (0.87 and 0.71±0.06 (with no OH scavenger) respectively). For ethyl vinyl ketone, primary carbonyl yields for formaldehyde (HCHO) and 2-oxobutanal have been determined to be 0.55 and 0.44 (Grosjean et al., 1996), and 0.37 and 0.49 (Kalalian et al., 2020) respectively,





displaying no clear preference for either fragmentation pathway. For α-β unsaturated ketones ($R_1CH=CHC(O)R_2$),
Grosjean and Grosjean (1999) measured the primary carbonyl yields from ozonolysis of 4-hexen-3-one to be:
acetaldehyde ($CH_3CHO$), 0.51±0.01, and 2-oxobutanal ($CH_3CH_2C(O)CHO$), 0.56±0.02, while Wang et al. (2015)
measured the primary carbonyl yields from ozonolysis of 3-methyl-3-buten-2-one ($CH_2=CR_1C(O)R_2$) to be:
diacetyl ($CH_3COCOCH_3$) 0.30±0.03, and HCHO 0.44±0.05, and from 3-methyl-3-penten-2-one
($R_1CH=CR_2C(O)R_3$), diacetyl 0.39±0.04 and $CH_3CHO$ 0.61±0.07. For ozonolysis of 2-enals, yields have been
reported for crotonaldehyde (2-butenal) ($CH_3CHO$ 0.42, glyoxal 0.47) (Grosjean and Grosjean, 1997) and trans-
2-hexenal (butanal 0.53, glyoxal 0.56) (Grosjean et al., 1996). For the atmospherically important isoprene
oxidation product methacrolein (2-methyl-prop-2-enal, MACR), Grosjean et al. (1993) measured yields of methyl
glyoxal of 0.58±0.06 and HCHO of 0.12±0.03. For 2-ethyl acrolein, the ethyl glyoxal yield has been measured to
be 0.14 by Grosjean et al. (1994), and 0.49±0.03 by O'Dwyer et al. (2010).

To summarise, the presence of a carbonyl group on a double bond appears to favour formation of the

opposing CI. However, this effect is neutralised to an extent by the presence of an alkyl substituent on the same
side of the double bond, e.g. in the case of 3-methyl-3-buten-2-one, methacrolein, and 2-ethyl acrolein. There
remain large uncertainties on the trends in these classes (it is noted that in some cases the sum of the measured
primary carbonyl yields is well below one). They clearly warrant further study, owing to the significance of these
classes of compounds in atmospheric chemistry (e.g. MACR and MVK from isoprene oxidation (Wennberg et al.,

2018)).

### 177   2.2.2   Enols / enol ethers

There has been very little experimental work on the atmospheric chemistry of enols due to difficulties in synthesis,
storage, and measurement of these compounds. However, two recent theoretical studies examined the ozonolysis
of enols. The first (Lei et al., 2020) on the simplest enol, vinyl alcohol (ethenol), suggested that formation of
$CH_2OO$ + HCOOH is strongly favoured (~78 %). The second (Wang et al., 2020), on the complex ketene-enol
species 4-hydroxy-1,3-butadien-1-one, also suggests that formation of HCOOH and the corresponding CI is
strongly favoured (84 %). By contrast, there have been several experimental studies on the product yields of the
reactions of enol ethers ($R_1\text{-}O\text{-}CR_2=CR_3R_4$) with ozone. Most studies (Thiault et al., 2002; Klotz et al., 2004;
Barnes et al., 2005; Zhou et al., 2006; Zhou, 2007; Al Mulla et al., 2010) have determined that the dominant POZ
decomposition channel yields the formate ($R_1\text{-}O\text{-}C(O)R_2$) and the corresponding CI ($R_3R_4COO$), with measured
yields of the formate ranging from 55 % - 89 % (see Spreadsheet S1). An exception to these studies is the work
of Grosjean and Grosjean (1997; 1999), which tended to find similar yields of the two primary carbonyl products.

### 189   2.2.3   Esters / acids

The primary carbonyl products of ozonolysis of the acrylate esters: methyl acrylate, ethyl acrylate, and methyl
methacrylate were studied by Bernard et al. (2010). Grosjean and Grosjean (1997) also studied methyl acrylate.
There is no clear evidence for a preferential route for POZ fragmentation in these studies (see Spreadsheet S1).
The primary carbonyl yields of vinyl acetate were measured to be 0.30±0.04 and 0.70±0.08 for HCHO and
$CH_3C(O)OC(O)H$ respectively by Al Mulla et al. (2010), and 0.20±0.06 and 0.97±0.08 by Picquet-Varrault et al.
(2010). These studies suggest a preference for formation of $CH_2OO$ and the acetate. There are only two
compounds reported for ozonolysis of α-β unsaturated acids: acrylic and methacrylic acid. For acrylic acid





ozonolysis in the presence of formic acid as an SCI scavenger, Al Mulla et al. (2010) measured yields of 1.48 ±
0.2 and < 0.1 for HCHO and HC(O)C(O)OH respectively, while in the absence of formic acid that group measured
a yield of HCHO of 0.95 (Viero, 2008). For methacrylic acid, Al Mulla et al. (2010) measured yields of 0.77 ±0.07
and 0.74 ±0.10 for HCHO and $CH_3C(O)C(O)OH$ respectively. It is difficult to rationalise these results: the acrylic
acid experiments suggest a preference for formation of the CI with the acid moiety, but the methacrylic acid
experiments suggest that the presence of a methyl group on the same side of the double bond as the acid reduces
this preference, in contrast to most other systems where methyl substitution increases the yield of that CI. This is
a recommended area for further study.
**2.2.4    Alcohols**
There are significant differences between measured primary carbonyl yields of α,β-unsaturated acyclic alcohols
between studies by Grosjean and Grosjean (1997), Le Person et al. (2009), O'Dwyer et al. (2010) and Kalalian et
al. (2020). This is likely owing to different experimental setups between groups, and the difficulty of quantitatively
measuring compounds with multiple oxygenated substituents. Overall the data in Spreadsheet S1 suggest that the
presence of a hydroxyl group in place of a hydrogen on the α-carbon may lead to a slight preference for CI
production on the other side of the double bond to the hydroxyl group.
**2.3    Conjugated alkenes**
The ozonolysis of conjugated alkenes leads to POZ with a vinyl substituent on the α-carbon. For non-symmetrical
conjugated alkenes, the measurement of primary carbonyl yields can only be used to determine the POZ
fragmentation if the relative contribution of reaction at each double bond to the overall reaction rate is known. For
ozonolysis of the atmospherically important biogenic alkene isoprene, the primary carbonyl yields recommended
by IUPAC (Atkinson et al., 2006; iupac-aeris.ipsl.fr, last accessed 6 December 2021) are: methyl vinyl ketone
(MVK), 0.17; methacrolein (MACR), 0.41; and HCHO 0.42. Based on reported product yields, the contribution
of reaction to each double bond to the overall rate has been estimated to be 0.6 for the terminal double bond and
0.4 for the substituted double bond (Nguyen et al., 2016; Jenkin et al., 2020). However, to the authors' knowledge
there has been no direct measurement of the reaction at each double bond, and this represents a significant
uncertainty in one of the most important atmospheric ozonolysis systems. Based on this assumption, and the
recommended yields of MVK and MACR, the formation of MACR+$CH_2OO$ is favoured over methacrolein oxide
(MACRO) + HCHO, and there is a slight preference for formation of methyl vinyl ketone oxide (MVKO) +
HCHO compared to MVK+$CH_2OO$. The MACR channel would suggest that the vinyl substituent is destabilising
compared to a hydrogen. The methyl group present in MVKO stabilises the CI, leading to a preference for this
channel. For symmetrical alkenes, the primary carbonyl yields should be directly representative of the POZ
fragmentation. For 1,3-butadiene, an acrolein yield of 51 – 52 % has been measured (Niki et al., 1983; Kramp and
Paulson, 2000), suggesting little preference for either POZ decomposition pathway, in contrast to the analogous
MACR channel in isoprene. Lewin et al. (2001) reported complementary carbonyl yields from ozonolysis at the
internal bond of (*E*) and (*Z*)-penta-1,3-diene and 5-methylhexa-1,3-diene, which all showed a preference for
formation of the unsaturated carbonyl (i.e. the saturated CI), suggesting that the vinyl group is less stabilising than
a methyl or isopropyl group, in agreement with the observations from isoprene.



## 2.4 Endocyclic alkenes

Decomposition of the POZ formed in the ozonolysis of endocyclic alkenes, leads to a molecule containing both the carbonyl oxide and carbonyl moieties. Thus for non-substituted cycloalkenes (e.g. cyclopentene) there is only one possible CI that can be formed (which can be in either the *E* or *Z* configuration). This means that there are no stable primary carbonyls formed and so the relative contributions of the POZ decomposition pathways cannot be inferred from measured primary carbonyl yields as they can for aliphatic compounds. Even a simple endocyclic system such as cyclohexene gives a complex range of gas-phase (Aschmann et al., 2003) and aerosol phase (Kalberer et al., 2000; Ziemann, 2002) products, which can be attributed to decomposition of both the *E* and *Z* forms of hexanal carbonyl oxide. However, the measured OH yields can be used to give an estimate of the amount of CI decomposing via the vinyl-hydroperoxide (VHP) pathway (see section 4.1). It is noted here that it has been proposed that alternative unimolecular pathways (that do not yield OH) are available to the CI formed from endocyclic alkenes (Chuong et al., 2004; Nguyen et al., 2009a; Long et al., 2019), but that these are only dominant for stabilised CI. Since the stabilised CI yield is low for endocyclic alkenes, at least up to $C_{10}$ (monoterpenes) (Chuong et al., 2004), measured OH yields should give a fair representation of the relative amount of CI decomposing via the VHP pathway). For non-substituted cycloalkenes, OH yields have been compiled by Calvert et al. (2000) covering cyclo-pentene, -hexene, -heptene, -octene and –decene from a number of research groups (Spreadsheet S2). There is some spread in the data but no clear evidence for favouring formation of *E* or *Z* CI, i.e. OH yields tend to centre around ~0.5. For substituted cycloalkenes, Atkinson et al. (1995) measured an OH yield of 0.90 for 1-methyl-1-cyclohexene, suggesting either that the dominant CI formed is the di-substituted CI (which will then undergo decomposition via the VHP pathway to yield OH), or that the mono-substituted CI is formed predominantly as the *syn* conformer. The former must be considered more likely based on the observed trends in aliphatic alkenes for favouring formation of the more substituted CI, and that there appears to be little preference for formation of *syn/anti*-CI from non-substituted endocyclic alkenes. 1-methyl-1-cyclohexene is particularly important from the point of view of atmospheric chemistry as an analogue for the abundant biogenic monoterpenes α-pinene and limonene. OH yields from α-pinene and limonene ozonolysis have been measured by a number of groups and are also generally high (0.64-0.91) (Cox et al. 2020), similar to 1-methyl-1-cyclohexene.

## 2.5 Exocyclic alkenes

For exocyclic alkenes in which the double bond is attached to the ring, e.g. β-pinene, the data suggests that POZ fragmentation strongly favours formation of the ring containing CI. For the monoterpene β-pinene, the mean measured yield of the $C_9$ carbonyl, nopinone, is 0.21 (Grosjean et al., 1993; Hakola et al., 1994; Rickard et al., 1999; Yu et al., 1999; Winterhalter et al., 2000; Hasson et al., 2001b; Lee et al., 2006; Ma and Marston, 2008), with theoretical work (Nguyen et al., 2009b) suggesting that some of this may be secondary and that the primary yield could be even lower. The other two compounds with a terminal double bond attached to the ring for which there are data are camphene (0.36 yield of $C_9$ carbonyl (Hakola et al., 1994; Hasson et al., 2001b)) and methylene cyclohexane (0.19 yield of $C_6$ carbonyl (Hasson et al., 2001b)). For the monoterpene sabinene, which has a terminal double bond attached to a $C_5$ and $C_7$ ring, the mean measured yield of the $C_9$ carbonyl, sabinaketone, is 0.44. This is considerably higher than from those compounds where the double bond is on a $C_6$ ring, probably demonstrating the impact of ring strain on the POZ fragmentation. The monoterpene terpinolene has a disubstituted double bond attached to a six membered ring. Reported yields of the ring containing carbonyl



(0.40±0.06 (Hakola et al., 1994); 0.40±0.08 (Reissell et al., 1999); 0.45 (Ma and Marston, 2009)) suggest yields
of the ring containing CI of 0.60 and 0.55 respectively; this assumes 100% reaction at the exocyclic double bond,
with Hakola et al. (1994) measuring a yield of ≤ 2 % of the dicarbonyl expected as a product (though by no means
the only one) from reaction at the endocyclic double bond. These CI yields are lower than for the exocyclic alkenes
with terminal double bonds, but are still considerably higher than most compounds which have a dimethyl
substitution on the double bond, for which acetone yields tend to be ~ 0.3. The presence of a ring clearly has a
different effect to simply having two alkyl groups attached to the double bond, leading to much higher yields of
the ring containing CI.

For alkenes with a vinyl group attached to a ring, there are data only for vinyl cyclohexane, and its aromatic

analogue styrene. These have similar yields for the ring containing carbonyl of 0.62 and 0.64 respectively
(Grosjean and Grosjean, 1997). There is no data for alkenes with double bonds more distant from a ring.

### 2.6    Yields of CI stereo-conformers

The formation of *syn/anti* conformers of CI in alkene ozonolysis was first discussed by Bauld et al. (1968), to
explain the observed *cis/trans* yields of the secondary ozonide formed from ozonolysis in the aqueous phase. Their
observations suggested that ozonolysis of *cis*-alkenes will predominantly form *anti*-CI, while for *trans*-alkenes
the predominance was less clear and appeared to be dependent on alkene structure. In the gas phase, but-2-ene is
the most studied system. Various experimental work has observed higher yields of OH from *trans*-but-2-ene
compared to *cis*-but-2-ene (see Spreadsheet S3). Assuming that only ($Z$)-CI decomposition yields OH (see Section
4.1), this implies a higher nascent ($Z$):($E$)-$CH_3CHOO$ ratio from decomposition of the POZ formed in *trans*-but-
2-ene ozonolysis. Orzechowska and Paulson (2002) measured a ratio of 1.62 for the OH yields from *trans/cis*-
but-2-ene. They observed a similar relationship for *trans/cis*-pent-2-ene and *trans/cis*-hex-3-ene, with OH yield
ratios determined as 1.80 and 1.51 respectively. Assuming that OH comes exclusively from ($Z$)-$CH_3CHOO$
implies a ($Z$):($E$)-RCHOO ratio of 0.60:0.40 – 0.64:0.36 for these three systems. Kroll et al. (2002) determined a
similar OH yield ratio for *trans/cis*-hex-3-ene, but using isotopically labelled hydrogen atoms demonstrated that
a fraction of this OH was not coming from the ($Z$)-CI. From their OH yield measurements, they inferred a ($Z$):($E$)-
$C_2H_5CHOO$ ratio of 50:50 for *trans*-3-hexene, and 20:80 for *cis*-3-hexene. Campos-Piñeda (2017) reported direct
measurements of the vinoxy radical formed in decomposition of *syn*-$CH_3CHOO$, from *cis*- and *trans*-but-2-ene
ozonolysis, inferring a yield of *syn*-$CH_3CHOO$ of ~0.5 from *trans*-but-2-ene and ~0.3 from *cis*-but-2-ene, broadly
in line with estimations from measured OH yields.

Early theoretical calculations considering the gas phase (Cremer, 1981a,b) suggested that ($Z$)-RCHOO is

likely to be formed in greater yield for small alkenes, but that ($E$)-RCHOO becomes more favoured in the
ozonolysis of large alkenes. Calculations by Rathman et al. (1999) suggested that ($Z$)-$CH_3CHOO$ should be
favoured in *trans*-but-2-ene ozonolysis, but that conversely ($E$)-$CH_3CHOO$ would be favoured in *cis*-but-2-ene
ozonolysis. Recent theoretical work (Watson, 2021) looking at POZ fragmentation for a series of disubstituted 2-
alkenes ($CH_3CH=CHR$), suggests formation of ($E$)-RCHOO will be strongly favoured in the ozonolysis of *cis*-
alkenes (87 % for *cis*-but-2-ene, increasing to 93 % for *cis*-2-hexene), while there is a roughly equal split from
ozonolysis of *trans*-alkenes. This is in qualitative agreement with the experimental work discussed above but
suggests a stronger preference than observed in the direct measurements of the vinoxy radical by Campos-Pineda
(2017). For tri-substituted alkenes, Watson (2021) finds a strong preference for formation of ($E$)-RCHOO on the



mono-substituted side of the double bond. For the $C_4$-CI formed in isoprene ozonolysis, theoretical calculations
have determined a relative split of 50:50 for the two conformers of MVKO (Kuwata et al., 2005), and 20:80 for
*syn*-MACRO:*anti*-MACRO (Kuwata and Valin, 2008). This is in qualitative agreement with the observed low
OH yield (0.08-0.13) from 1,3-butadiene (Atkinson and Aschmann, 1993; Kramp and Paulson, 2000) if it is
assumed that decomposition of *syn*-MACRO will have a high OH yield whereas *anti*-MACRO will not yield OH.
To the authors' knowledge there is no other information on the relative yields of *syn*/*anti*-$R_1R_2COO$ (where $R_1 \neq$
$R_2$).
**2.7    POZ ring opening to a biradical**
In addition to direct CI + carbonyl formation from the POZ, the possibility exists of ring opening of the POZ to a
singlet alkoxy-peroxy biradical ($>C(O^\bullet)-C(OO^\bullet)<$) (O'Neal and Blumstein, 1973; Olzmann et al., 1997; Anglada
et al., 1999; Fenske et al., 2000; Nguyen et al., 2015; Pfeifle et al., 2018) (Figure 4). In addition to re-closing the
ring to the POZ or decomposing to CI + carbonyl, this alkoxy-peroxy biradical can migrate an H-atom from the
alkoxy-bearing carbon, forming a carbonyl hydroperoxide (–C(=O)-C(OOH)<); this pathway is only possible if
the alkene has a vinylic H-atom. The carbonyl hydroperoxide formed has a high energy content, over 100 kcal
mol$^{-1}$, and can eliminate an OH radical, forming a α-carbonyl-alkoxy radical that rapidly decomposes to an acyl
radical and a carbonyl. This pathway has been invoked in theoretical studies as the main source of OH in the
ozonolysis of ethene (in which OH cannot be formed via a VHP) (Nguyen et al., 2015; Pfeifle et al., 2018), and
is expected to contribute somewhat to OH formation in other alkenes, though this has not yet been investigated
experimentally or theoretically. Alternative proposed sources of OH in ethene ozonolysis all involve the $CH_2OO$
Criegee intermediate. However, theory has shown that direct OH formation from $CH_2OO$ by a 1,3-H-migration
involves too high a barrier (e.g. Nguyen et al., 2015; Pfeifle et al., 2018), while OH elimination from the hot
formic acid formed in the 1,3-ring closure (see Section 4.2) is not competitive against formation of $H_2O + CO$ and
$H_2 + CO_2$, as also borne out by HCOOH pyrolysis experiments (Chang et al., 2007; Vichietti et al., 2017). The
carbonyl hydroperoxide route thus resolves an apparent discrepancy between ethene ozonolysis experiments,
which observe significant OH yields, and experiments (Stone et al., 2018) and theoretical work (Nguyen et al.,
2015; Pfeifle et al. 2018), which indicate very little OH formation from $CH_2OO$. Pfeifle et al. (2018) calculated a
yield of 12.3 % for the carbonyl-hydroperoxide in ethene ozonolysis, while Nguyen et al. (2015) obtained 13 %,
both at the low end of the current IUPAC recommended OH yield (0.17±0.05) for the reaction (Cox et al., 2020).

$C_2H_4$          POZ                                CHP
**Figure 4. The carbonyl hydroperoxide (CHP) decomposition pathway for ethene ozonolysis**





**2.8** **Protocol Rules for POZ fragmentation**
**2.8.1** **POZ fragmentation**
A group contribution approach was designed to estimate POZ fragmentation yields. The approach assumes that
the branching ratio for the two possible fragmentations of the POZ depends on the substituents of the
$R_{1a}(R_{1b})C=C(R_{2b})R_{2a}$ parent alkene. The general form of the relationship is given by:

$$Y_{CI1} = \frac{(F_{1a}+F_{1b})-(F_{2a}+F_{2b})+1}{2} = 1 - Y_{CI2} \qquad (E1)$$
where $Y_{CIi}$ is the CI production yield on the $i^{th}$ carbon and $F_R$ are the contributions for the 4 substituents on the
C=C bond. The set of $F_R$ values is developed based on the observed primary carbonyl yields (Supplementary
Section S1 and Spreadsheet S1) and are based on a least squares fit to a relevant dataset of alkenes for each
substituent (Figures S1-S5).
For a vinyl group, $F$ is constrained to fit the IUPAC recommended yields of MVK and MACR from
isoprene ozonolysis, assuming that ozone reacts 60% at the terminal double bond and 40% at the substituted
double bond (Nguyen et al., 2016; Jenkin et al., 2020). The presence of a carbonyl group adjacent to the double
bond appears to strongly favour formation of the opposing CI in the case of MVK (i.e. -C(=O)H). However, this
is not the case for alkenes with the structure -C(=O)R in the database, for which there appears to be no clear
preference for formation of either CI, with a fit to the data yielding a slightly positive $F$ value of 0.127. The
strongest negative effect (i.e. leading to formation of the carbonyl containing the functional group) observed in
the database is for enol ethers (-OR), giving an $F$ value of -0.655. This is assumed to also be the same case for
enols (-OH) based on the theoretical calculations of Lei et al. (2020) and Wang et al. (2020), and for vinyl esters
(-OC(=O)R), based on the observed values for vinyl acetate. By contrast, an acrylate ester (-C(=O)-O-) substituent
adjacent to the double bond does not appear to have a strong effect on fragmentation, and $F = 0$ is used. Similarly,
the trend from the two unsaturated acids reported is unclear, and $F = 0$ is also used here. An OH group on the
alpha carbon appears to slightly decrease $Y_{CI}$ compared to an H atom, but the data is currently too limited to
recommend a group additivity value, so the OH group is treated as an H atom, i.e. $F_{-CH2OH} = F_{-CH3}$. More distant
oxygenated groups are not considered. The available data for exocyclic alkenes with the double bond attached to
the ring is not able to take into account the effect of multiple rings, with $F_{=ring}$ being determined from only
exocyclic alkenes with $C_6$ rings (β-pinene, methylene cyclohexane, and terpinolene). For rings with a vinyl group
attached, $F_{(C6)ring}$ is determined only from $C_6$ rings, i.e. styrene and vinylcyclohexane. Endocyclic alkenes are
assumed to follow the same fragmentation patterns as acyclic alkenes. For example, cyclohexene is considered to
have the structure >CH₂CH₂CH=CHCH₂CH₂<, 1-methyl cyclohexene >CH₂CH₂C(CH₃)=CHCH₂CH₂< etc.
The group contribution value, $F$, is then used in Eq. (1) to determine the yield of $CI_1$ (defined as having
substituents 1a and 1b) from the general structure $R_{1a}(R_{1b})C=C(R_{2b})R_{2a}$. Generally, the measurement of the larger
primary carbonyl was used to determine the primary carbonyl and CI yields. This is because in some cases, the
smaller carbonyl can be formed as a decomposition product of the larger CI and hence is not a true primary
carbonyl yield.



**Table 1. Group contribution values (F) for various substituents**

| Group | Value | Alkenes used for fit |
|---|---|---|
| =ring | + 0.62 | β-pinene, methylene cyclohexane, terpinolene |
| -CH$_3$ | + 0.218 | propene, 2-methyl butene, 2-methyl-but-2-ene |
| -C(=O)R | + 0.127 | 2-ethylacrolein, ethyl vinyl ketone, 4-hexen-3-one, 3-methyl-3-buten-2-one, 3-methyl-3-penten-2-one, 2-butenal, trans-2-hexenal |
| -CH$_2$CH$_3$ | + 0.107 | but-1-ene, 2-methyl-but-1-ene, 2-ethyl-but-1-ene, 2,2-dimethyl-hex-2-ene |
| -H | 0 | By definition |
| -COOH, -C(=O)-O-R, | 0 | Acids and acrylate esters, see spreadsheet S1 |
| -CH$_2$CH$_2$R | 0 | pent-1-ene, hex-1-ene, hept-1-ene, oct-1-ene, dec-1-ene, 2-methyl-pent-1-ene |
| -CHR$_1$R$_2$ | - 0.069 | 3-methyl-but-1-ene, 3-methyl-pent-1-ene, 2,3-dimethyl-but-1-ene, 2,4-dimethyl-pent-2-ene, 2,3,4-trimethyl-pent-2-ene, 3-methyl-2-isopropyl-but-1-ene |
| -(C$_6$)ring | - 0.25 | styrene, vinyl cyclohexane |
| -vinyl | - 0.28 | isoprene |
| -CR$_1$R$_2$R$_3$ | - 0.386 | 2,3,3-trimethyl-but-1-ene, 2,4,4-trimethyl-pent-2-ene, 2,2-dimethyl-hex-3-ene, 3,3-dimethyl-but-1-ene |
| -OR, -OH, -(-OC(=O)R | - 0.655 | methyl vinyl ether, ethyl vinyl ether, propyl vinyl ether, butyl vinyl ether, ethyl propenyl ether |

### 2.8.2 *E / Z* conformer yields
In light of the current paucity of experimental and/or theoretical information on the relative yields, an equal 0.5:0.5
yield is assigned as a default value for *(E)/(Z)* isomers for all asymmetrical CI. The following two exceptions are
nevertheless considered. For acyclic *cis*-RCH=CHR parent alkenes, a relative yield of 0.7:0.3 is set for (*E*):(*Z*) CI.
For conjugated structures, formation of (E)/(Z)- >C=C(R)-CHOO is assumed to be in a ratio of 0.8:0.2, based on
the work of Kuwata et al. (2005) and Kuwata and Valin (2008).
### 2.8.3 Carbonyl-hydroperoxide route
While there is little information available on the stepwise carbonyl-hydroperoxide POZ decomposition
mechanism, it is needed to account for the radical yields observed in the ozonolysis of ethene as discussed above.
There is no reason to assume it will not occur more generally for any alkenes with vinylic H-atom(s), though
perhaps with different fates of the intermediate biradical or carbonyl hydroperoxide. Currently this channel is only
included for the ethene-ozone reaction, for which it is assumed that 0.12 of the ethene-ozone reaction forms the
biradical intermediate, rather than the CI + carbonyl, using the contribution calculated for the carbonyl
hydroperoxide channel by Pfeifle et al. (2018). When more general data become available, assuming the channel
is active for other systems, the protocol will be updated. The general structure of such a scheme might be: the
POZ is assumed to break either of the O-O bonds with equal probability, forming one of two possible biradicals.
If there is an available vinyl α-hydrogen, it is assumed that the H-shift to the peroxy radical occurs, forming the
carbonyl-hydroperoxide (R$_1$R$_2$C(OOH)C(=O)R$_3$), followed by loss of OH and scission of the C-C bond to yield
the stable product R$_1$R$_2$C=O and the radical R$_3$C$^•$=O. If there is no available α-hydrogen, the biradical is assumed
to yield the CI and carbonyl, either by C–C fragmentation or recyclisation to the POZ.
## 3 Stabilisation of the Criegee Intermediate
### 3.1 Excited vs. stabilised CI
Following decomposition of the primary ozonide, CI are formed with a broad range of internal energies (e.g.
Drozd et al., 2011). Consequently, it is often useful to consider the mean energy of a population of CI. Those



generated with a high internal energy, allowing prompt chemical reactions, are called excited, or chemically
activated CI (CI*). Those without enough internal energy to undergo prompt decomposition are considered to be
'stabilised' CI (SCI). Additionally, CI* can be collisionally stabilised. This has been demonstrated by
experimental work showing that SCI yields are pressure dependent (Drozd et al., 2011, Hakala and Donahue,
2016; 2018). Note that this pressure dependence is moderate, and across the range of relevant atmospheric
pressures not of primary concern; we base our analysis on the available data near 1 atm.

## 3.2    SCI Yield

The total SCI yield for a given alkene is the sum of the fraction of the nascent CI population that is formed
stabilised, plus the fraction of CI* that is collisionally stabilised. The fate of the CI* is a competition between
prompt unimolecular decay and collisional stabilisation, with the CI* having a lifetime on the order of
nanoseconds against either of these processes (e.g. Drozd et al., 2017; Stephenson and Lester, 2020). Most alkenes
will form a number of different CI*, each with different lifetimes against unimolecular decay and collisional
stabilisation. The rate of collisional stabilisation of a given CI* is dependent on the frequency of collisions (and
hence pressure), and the efficiency of energy loss to the bath gas. The rate of unimolecular decay of a given CI*
depends on: (i) the energy of the CI* when formed, (ii) the activation energy for the most facile decay process /
the energy required for tunnelling, and (iii) the relative density of states of the reactants and transition state, i.e.
the entropy of the reaction. The dominant unimolecular decay mechanism is dependent on the structure of the CI;
these mechanisms are discussed in Section 5.

Larger CI* will tend to be stabilised to a greater extent due to a greater density of states distributing the

excess internal energy over a greater number of modes and so reducing the rate of unimolecular decay (Drozd and
Donahue, 2011; Stephenson and Lester, 2020). However, as the size of the CI increases relative to the carbonyl
co-product formed in POZ decomposition, the fraction of the energy taken by the CI from the POZ will increase
(assuming the energy has time to become equally distributed throughout the CI), decreasing the mean energy of
the nascent CI population and hence the fraction of CI* with enough energy to undergo unimolecular decay
(Fenske et al., 2000; Newland et al., 2020). This will lead to greater stabilisation. For endocyclic alkenes,
decomposition of the POZ produces a single molecule containing both the carbonyl and carbonyl oxide moieties.
Such CI have a high initial energy, with no energy lost from the POZ decomposition to relative motion of the
fragments, and thus require many collisions to be quenched (Vereecken and Francisco, 2012). Consequently,
endocyclic alkenes with $\leq C_7$ have little stabilisation (Hatakeyama et al., 1984; Campos-Pineda and Zhang, 2017;
Drozd and Donahue, 2011). For the endocyclic $C_{10}$ monoterpenes α-pinene and limonene, total SCI yields have
been measured to be 0.13-0.22 (Hatakeyama et al., 1984; Taipale et al., 2014; Sipilä et al., 2014; Newland et al.,
2018) and 0.23-0.27 (Sipilä et al., 2014; Newland et al., 2018) respectively. For the $C_{15}$ sesquiterpene β-
caryophyllene, a total SCI yield (including from decomposition of the stabilised POZ) of 0.74 was calculated by
Nguyen et al. (2009), with a value of > 0.6 determined experimentally (Winterhalter et al., 2009).

Total SCI yields have been measured experimentally for many alkene-ozone systems. These are generally

determined indirectly, by performing ozonolysis experiments in the presence of an SCI scavenging species (e.g.
$H_2O$, $SO_2$, hexafluoroacetone). Measurements of scavenger removal, or formation of products from the SCI +
scavenger reaction, are used to determine the SCI yield. Yields measured in such a way must be considered to be
lower limits since, under most experimental conditions, a significant fraction of the SCI may undergo





unimolecular decomposition based on recently reported fast SCI decomposition rates (e.g. Newland et al., 2015;
Vereecken et al., 2017; Newland et al., 2018). The choice of scavenger species is also important. In some older
experimental studies, water was used as an SCI scavenger, with $H_2O_2$ (e.g. Hasson et al., 2001a) or hydroxymethyl
hydroperoxide (HMHP, e.g. Hasson et al, 2001a; Neeb et al., 1997) being the detected reaction products. For
mono-substituted (E)-SCI, or for $CH_2OO$, this may be a reasonable assumption, with $k_{(H2O+SCI)}[H_2O]/k_{(decomp.)} \sim$
$10^2$ - $10^3$ at $[H_2O]$ = $5 \times 10^{17}$ cm$^{-3}$ (e.g. Vereecken et al., 2017). However, for (Z)-SCI, $k_{(H2O+SCI)}[H_2O]/k_{(decomp.)} \sim$
$10^{-2}$ - $10^{-1}$, i.e. the majority of the SCI will not be scavenged by $H_2O$.

### 3.3 Protocol Rules for CI Stabilisation

The relationship between stabilisation of the CI* and size of the carbonyl co-product has been studied for $CH_2OO$
and $(CH_3)_2COO$ by Newland et al. (2020) (Figure 5). For $CH_2OO$ this relationship might be expected to represent
a minimum for CI* that primarily decay via the 1,3 ring closure pathway (i.e. *anti*-CI*, see Section 4.2), since
larger CI* will have a slower decay rate due to a greater density of states. Similarly, the trend for $(CH_3)_2COO$ can
be assumed to be close to a minimum for CI* that primarily undergo the 1,4 vinylhydroperoxide (VHP)
decomposition pathway (see Section 4.1), with only *syn*-$CH_3CHOO$ likely to have a lower density of states (and
therefore faster decomposition) (Stephenson and Lester, 2020). With no further data available, the stabilisation
trend of $CH_2OO$ is used for CI* that decompose via 1,3 ring closure, while that of $(CH_3)_2COO$ is used for CI* that
decay via the 1,4 vinylhydroperoxide pathway. For other pathways, such as the 1,5-ring closure to a dioxole (see
Section 4.4), important in isoprene ozonolysis, no information is available. CI* with a vinyl group *syn* to the
terminal oxygen of the carbonyl oxide are considered as *syn*-CI for the purposes of calculating stabilisation in the
protocol.
An extension of Equation E7 in Newland et al. (2020) is used to estimate the CI stabilisation $S$:

$$S = 1 - \left[ \left( \frac{A_{CI}}{A_{tot}} \right) \times F \times z_{path} \right] \qquad (E2)$$
where $A_{CI}$ is the total number of non-hydrogen atoms in the CI*, $A_{tot}$ is the total number of non-hydrogen atoms in
the POZ, and $F_{VHP}$ and $F_{13RC}$ are values determined for $CH_2OO$ and $(CH_3)_2COO$ based on the SCI yields for their
symmetrical parent alkenes ethene and 2,3-dimethylbut-2-ene, respectively. For $CH_2OO$ this is 0.95 and for
$(CH_3)_2COO$ it is 1.24 (Newland et al., 2020). In this work, an additional term, $z_{path}$, is included to take into account
the observed / predicted increased stabilisation of CI* with size. For CI* that decay via the 1,3 ring closure
pathway, $z_{13RC}$ is defined as $x / (A_{CI} + (x - A_{CH2OO}))$, where $A_{CH2OO}$ is the total number of non-hydrogen atoms in
$CH_2OO$ (i.e. 3), and $x$ is an adjustable parameter. For CI* that decay via the 1,4 H-shift, $z_{VHP}$ is defined as $x / (A_{CI}$
$+ (x - A_{(CH3)2COO}))$, where $A_{(CH3)2COO}$ = 5. In both terms, $x$ = 5, and has been optimized to improve the fit between
measured and calculated total SCI yields of larger alkenes (Newland et al., 2020).
Figure 5 shows the measured CI* stabilisation for $CH_2OO$ and $(CH_3)_2COO$ as a function of the total
energy taken from the POZ by the CI*, from Newland et al. (2020). Fits to the measured data are calculated using
Eq. (2). Also shown are the calculated stabilisation trends for (E)- and (Z)-$CH_3CHOO$ and nopinone oxide (the $C_9$
CI* formed in β-pinene ozonolysis). Figure 5 shows that stabilisation of *E*-CI* is predicted to be considerably
greater than for *Z*-CI* when formed with the same energy. For $CH_3CHOO$ it is noted that very little (0.11)
stabilisation of (Z)-$CH_3CHOO$* is predicted when produced from but-2-ene ozonolysis (fraction of total energy





$= A_{CI}/A_{tot} = 4/7 = 0.57$), whereas a much greater stabilisation of $(E)$-CH$_3$CHOO* is predicted. Using the $E/Z$-
RCHOO yields given in Section 2.8.2 for *cis* and *trans* alkenes, and the trends presented in Figure 5, then a total
SCI yield of 0.33 for *trans*-but-2-ene and 0.42 for *cis*-but-2-ene is calculated, in good qualitative agreement with
the relationship observed in Newland et al. (2015). The calculated values for nopinone oxide demonstrate the
decreasing sensitivity of CI* stabilisation to the co-product size as the size of the CI* increases.

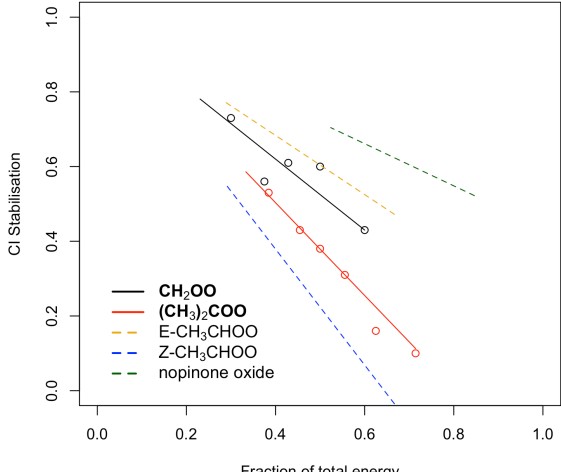


**Figure 5. Dependence of CI\* stabilisation on the fraction of the total energy taken from the POZ. Black (CH$_2$OO) and**
**red ((CH$_3$)$_2$COO) points, measurements taken from Newland et al. (2020). Solid and dashed lines, fits calculated using**
**Eq. (2).**
For endocyclic alkenes, an empirically derived sigmoid fit (Supplementary Section S2: Equation S1; Figure
S6) is applied to the very limited dataset that shows $Y_{SCI} \approx 0$ for $C \leq 7$, $Y_{SCI} \approx 0.2$ for monoterpenes, and $Y_{SCI} \approx$
0.74 for sesquiterpenes.
**4**    **Unimolecular reactions of CI\* and SCI**
CI can undergo unimolecular isomerisation / decomposition. The unimolecular pathways available to SCI are
assumed to be the same as those available to CI* (although it is noted that there is little evidence to back up this
assumption). However, while for CI* these processes are prompt, occurring on a timescale of 10$^{-9}$ s (Drozd et al.,
2017), for SCI they occur at a range of rates such that their competition with atmospheric bimolecular reactions
needs to be considered. A wide range of unimolecular isomerisation / decomposition pathways have been
characterised for CI, but only two of these are believed to be important for saturated CI under atmospheric
boundary layer conditions (Vereecken et al., 2017): a 1,4 H-migration, i.e. the vinylhydroperoxide pathway, and
a 1,3 ring closure, i.e. the hot acid / ester pathway (Figure 6). If the vinylhydroperoxide pathway is available, then
this will always be the dominant decomposition pathway as it is the energetically most facile, with only a slight
entropic disadvantage compared to the 1,3 ring closure (Vereecken et al., 2017). Unsaturated CI have some
additional pathways available (see Section 4.4).
Experimentally determined decomposition rates are available only for a limited number of SCI. Early
estimates were considerably slower than more recent experimental evidence. Vereecken et al. (2017) recently





published an extensive SAR providing temperature dependent unimolecular rates and mechanisms for a wide
range of SCI structures based on theoretical calculations tied to experimental work as well as group additivity
relations.

**Figure 6. Available pathways for a CI with a hydrogen atom available in beta position to the carbonyl oxide. From top**
**to bottom, the available pathways are the stabilisation (stab.) pathway, the vinylhydroperoxide (VHP) pathway and**
**the 1,3 ring closure (hot acid/ester) pathway.**
**4.1    Vinylhydroperoxide (VHP) pathway**
A CI with a β-hydrogen atom in a *syn* orientation to the terminal oxygen atom of the carbonyl oxide can isomerise
to form a vinylhydroperoxide via a 5-membered transition cycle (Figure 6). This route is therefore available to
monosubstituted (*Z*)-CIs and disubstituted CIs. The VHP formed has a short lifetime and promptly or thermally
decomposes to form an OH radical and a β-acylalkyl (vinoxy) radical, in some cases with a small yield of β-acyl-
alcohols (Taatjes et al., 2016; Kuwata et al., 2018). The OH radicals are thus formed on a short time scale (e.g.
Drozd et al., 2017) directly from the VHP decomposition. The β-acylalkyl radical reacts with $O_2$ to form a β-
acylperoxy radical. On a longer timescale, the subsequent chemistry of this peroxy radical can yield further $HO_2$
and OH radicals (e.g. Nguyen et al., 2016).

The best studied system that follows the 1,4 H-shift pathway is stabilised $(CH_3)_2COO$. Experimentally

derived rates are fast (300 – 1000 s$^{-1}$) (Berndt et al., 2014b; Newland et al., 2015; Chhantyal-Pun et al., 2016;
Smith et al., 2016). The experimental evidence also shows a strong temperature dependence, with measured rates
varying from 269 s$^{-1}$ at 283 K to 916 s$^{-1}$ at 323 K (Smith et al., 2016). This is in good agreement with the SAR of
Vereecken et al. (2017) which shows that the rate of decomposition of saturated SCI is fastest (*ca.* 500 s$^{-1}$) for
those SCI with access to the VHP route. This SAR shows that the rate is slowed by more than an order of
magnitude when only one H atom is available on the α-carbon and that the rates are also affected by the *anti*
substituent, with the presence of a vinyl group reducing rates by an order of magnitude, and the presence of a
carbonyl group reducing rates by two orders of magnitude.





This pathway may not be available to certain CI structures even though there is an available hydrogen on

the α-carbon. This is the case for the bicyclic $C_9$ CI formed in ozonolysis of the monoterpene β-pinene, with the
terminal oxygen facing the four membered ring. Calculations have shown that formation of the vinyl
hydroperoxide is not possible for this CI due to the strain it would put on the ring, and so the dominant
decomposition pathway is 1,3 ring closure (Nguyen et al., 2009b). This has also been shown to be the case for the
cyclic $C_9$ CI formed facing the three membered ring in the ozonolysis of sabinene (Almatarneh et al., 2019).
**4.2      1,3 ring closure**
For monosubstituted (E)-CI and $CH_2OO$ (see Section 5.3), decomposition via a VHP is not available. Instead
unimolecular reaction proceeds predominantly via a 1,3 ring closure, with typical rates $\leq 10^2$ s$^{-1}$ (Vereecken et al.
2017), to a chemically activated dioxirane species (Figure 6). This breaks the weak O-O bond giving a singlet bis-
oxy radical (Wadt and Goddard, 1975; Herron and Huie, 1977; 1978). Various pathways have been proposed for
the subsequent chemistry of this species based on observed product distributions (Chen et al., 2002). This pathway
has been characterised best for $CH_2OO$ (Section 5.3). The dioxirane is thought to rearrange to a 'hot' acid / ester,
which can undergo decomposition to yield a range of products. As the size of the CI increases, the hot acid / ester
is predicted to be more likely to be collisionally stabilised (Vereecken and Francisco, 2012).

There have been very few experimental studies to date on the products of isomerisation / decomposition

of (E)-RCHOO. This is challenging experimentally as (E)-RCHOO will always be formed as a partner with (Z)-
RCHOO. The most studied (E)-CI is (E)-$CH_3CHOO$, with observed products from *cis/trans*-but-2-ene ozonolysis
(which yields (E)- and (Z)-$CH_3CHOO$ as the CI products) of HCHO, $CH_3COOH$, $CH_3OH$, $CH_4$, CHOCHO,
ketene, CO and $CO_2$ (e.g. Tuazon et al., 1997; Grosjean et al., 1994). With the exception of glyoxal, these can all
be rationalised as decomposition products of 'hot' (E)-$CH_3CHOO$ via various pathways (Reactions R1 – R5). The
relative proportion of each channel is based on the reported yields in Tuazon et al. (1997), except for $CH_3COOH$,
from Grosjean et al. (1994), although it is noted that $CH_3COOH$ may be a product of $CH_3CHOO$ + water vapour
in their experimental setup.

$E$-$CH_3CHOO$ → $CH_4 + CO_2$            25 %         (R1)

$CH_3 + CO_2 + H$            25 %         (R2)

$CH_3OH + CO$               15 %         (R3)

$H_2CCO + H_2O$             10 %         (R4)

$CH_3COOH$                   20 %         (R5)


For $R_1R_2COO$ decomposition via 1,3 ring closure, products are formed via a 'hot' ester. There has been very little
work on the relative contribution of decomposition channels and stabilisation for these species. For example, there
is no experimental work to validate the predicted trend of increasing stabilisation of the hot acid / ester with size,
or at what size this becomes important. For the large terpenoid compounds β-pinene (Nguyen et al., 2009b) and
β-caryophyllene (Nguyen et al., 2009a), the acids/lactones formed from isomerisation of the $C_9$-dioxirane have
been predicted to be fully stabilised.




### 4.3 CH₂OO


CH$_2$OO also follows the 1,3-ring closure pathway but is considered separately here as it has been the subject of a
considerable body of work. Experimentally reported products from CH$_2$OO decomposition include: CO$_2$, CO, H$_2$,
OH, HO$_2$, H$_2$O, and HCOOH (e.g. Calvert et al., 2000). Recent theoretical (Nguyen et al., 2015; Stone et al., 2018;
Peltola et al., 2020) works suggest that the only reaction pathway of the bis-oxy radical important under
tropospheric conditions is isomerisation to 'hot' formic acid, followed by decomposition to either H$_2$ + CO$_2$ or
H$_2$O + CO, in agreement with experimental and theoretical work on acid pyrolysis experiments (Chang et al.,
2007; Vichietti et al., 2017). Due to the large excess energy and its small size, very little of the hot acid is stabilised,
with measured HCOOH yields from ethene ozonolysis < 5% (Calvert et al., 2000) (and the latter may be due to
bimolecular reactions of SCI rather than stabilisation of the hot acid). Stone et al. (2018) and Peltola et al. (2020)
considered the decomposition of stabilised CH$_2$OO using master equation simulations, determining the major
decomposition channel to be H$_2$+CO$_2$ (64 % and 61% respectively), with the H$_2$O+CO accounting for the
remainder (36%) in Stone et al. (2018), while Peltola et al. (2020) also found a small contribution (~8%) from the
OH+HCO channel. It is noted that previous experimental work on ethene ozonolysis (Su et al., 1980; Horie et al.,
1991; Neeb et al., 1998) has generally inferred a preference for the H$_2$O+CO channel. This may be due to different
pathways being followed by the dioxiranes formed from the excited CH$_2$OO produced in the ozonolysis reaction
compared to those formed from stabilised CH$_2$OO, as suggested by work on larger systems (Nguyen et al., 2009a;
2009b), and in the calculations of Nguyen et al. (2015) on excited CH$_2$OO decomposition in ethene ozonolysis. A
decomposition pathway to HCO + OH, proposed as the source of observed OH yields of 8-15 % in earlier
experimental studies on the ozonolysis of ethene (Gutbrod et al., 1997; Rickard et al., 1999; Kroll et al., 2001;
Alam et al., 2011) and larger alkenes (Kroll et al., 2002), has recently been determined experimentally to be
negligible (Stone et al., 2018), accounting for less than 2 % of the overall decay. This is in agreement with earlier
theoretical work (Olzmann et al., 1997; Nguyen et al., 2015) suggesting negligible OH yields from ethene
ozonolysis. This apparent discrepancy between experiment and theory can be reconciled by invoking the
possibility of OH formation via the carbonyl-hydroperoxide channel in the POZ decomposition, as discussed in
Section 2.7.
The unimolecular decomposition rate of stabilised CH$_2$OO has been experimentally determined to be very
slow (<12 s$^{-1}$) (Berndt et al., 2015; Chhantyal-Pun et al., 2015; Newland et al., 2015; Stone et al., 2018; Peltola et
al., 2020), with a current recommendation by IUPAC of ≤ 0.2 s$^{-1}$ at 1 bar and 298 K (Cox et al., 2020). Even at
the upper end of these estimates, decomposition is a negligible atmospheric fate for stabilised CH$_2$OO compared
to reaction with water vapour.

### 4.4 Unimolecular reactions of unsaturated CI


The ozonolysis of conjugated alkenes proceeds via the same initial POZ mechanism as non-conjugated systems,
but decomposition of the POZ leads to the formation of unsaturated CI and/or carbonyls. While many of the
characteristics of the chemistry are expected to be similar, the theoretical work of Kuwata et al., (2005), Kuwata
and Valin (2008), and Vereecken et al. (2017) has shown some important differences. Specifically, additional
unimolecular decomposition channels (Figure 7 and Figure 8) become available, which in some cases are faster
than the 1,4 H-shift channel.





**Figure 7. Dominant unimolecular decomposition routes available to unsaturated CI with the terminal oxygen *syn* to an α or β vinyl group. Pathways available if terminal oxygen is *anti* to a vinyl group are the same as for saturated CI. For 1,5-ring closure see Figure 8.**

If the vinyl group of an unsaturated CI is *anti* to the terminal oxygen of the carbonyl oxide, then the molecule will follow one of the two routes available to saturated CI, but with a rate affected by the presence of the double bond. However, if the vinyl group is *syn* to the terminal oxygen, alternative mechanisms of decomposition are available. 1,4 and 1,6-allyl H-migration (for the vinyl group being in β or α position respectively), are available if an H atom is present on the α or γ carbon. These pathways lead to similar products to 1,4-alkyl H-migration, with a vinylhydroperoxide intermediate decomposing to give OH and one of two possible unsaturated peroxy radicals. If no H-atom is available for (*Z*)-β-unsaturated CI then they follow the 1,3-ring closure channel with SCI decomposition rates $\leq 1$ s$^{-1}$. The rates of the 1,6-allyl H-migration channel for SCI are of the order of $10^6$ s$^{-1}$, while 1,4-allyl H-migration of SCI has rates ranging from $10^1 - 10^4$ s$^{-1}$ depending on other substituents (Vereecken et al. 2017).

For CI with the carbonyl oxide *syn* to an α vinyl group, and without an available hydrogen on the α carbon, then the dominant decomposition mechanism is 1,5 ring closure, originally proposed by Kuwata et al. (2005) (Figure 8). This forms an intermediate dioxole species with a five membered ring. This is predicted to have high internal energy and to break the O-O bond, leading to an epoxy carbonyl, or, if $R_4$ = H, to a dicarbonyl (Kuwata 2005). The dicarbonyl has been predicted to undergo further prompt decomposition via various possible unimolecular channels, some of which appear to yield OH (Barber et al., 2018). Based on the stable product distribution from *anti*-MVKO decay, the decomposition of the dicarbonyl has been determined to be predominantly via C-C cleavage leading to two radicals (acetyl and vinoxy radicals in the case of *anti*-MVKO) (Vansco et al., 2020). These radicals will add O$_2$ leading to RO$_2$ radicals which may undergo further decomposition if formed chemically excited, ultimately to HCHO + OH + CO in both cases (Carr et al., 2011; Weidman et al., 2018; Vansco et al., 2020). For *syn*-MACRO, Vansco et al. (2020) determine a pathway via a dioxole analogous to that just described, leading to formyl and 2-methyl vinoxy radicals, the latter of which could ultimately yield CH$_3$CHO + OH + CO. However, this accounts for only about half of the decomposition of the dicarbonyl, with the other half leading to acrolein via an unidentified unimolecular process. It is noted that Barber



et al. (2018) and Vansco et al. (2020) did not consider the epoxide isomerisation pathway for the dioxole. The
calculated unimolecular decay rates for the dioxole forming pathways from *syn*-MACRO and *anti*-MVKO are
fast, Vereecken et al. (2017) reported a rate of 13,400 s$^{-1}$, while Barber et al. (2018) reported a somewhat slower
rate for *anti*-MVKO of 2140 s$^{-1}$. Decay of stabilized *syn*-MVKO is relatively slow at 33 – 50 s$^{-1}$ (Vereecken et al.,
2017; Barber et al., 2018) making it a potentially important bimolecular reaction partner in the atmosphere.


**Figure 8. 1,5-ring closure: dominant unimolecular pathway for unsaturated CI with the terminal oxygen *syn* to an α vinyl group and R$_4$ is not a carbon with an abstractable hydrogen.**

**4.5    Protocol Rules for CI decomposition**
For unimolecular decomposition of CI, the SAR of Vereecken et al. (2017) is used to determine decomposition
pathways and rates (for SCI). The products from each decomposition pathway are given in Table 2, where any
secondary reactions such as recombination with $O_2$ are already accounted for. The vinylhydroperoxide pathway
is assumed to lead exclusively to a β-oxo alkyl radical and OH. For decomposition via 1,3 ring closure, the hot
acid / ester formed is considered to decompose via one of the three major pathways determined for (*E*)-RCHOO:
RH + $CO_2$ (40%), ROH + CO (20%), R + $HO_2$ + $CO_2$ (40%), based on the observed product yields from *cis* and
*trans* but-2-ene experiments by Tuazon et al. (1997). While it is noted that Grosjean et al. (1994) observed a
$CH_3COOH$ yield of ~ 20 %, this could also be a product of $CH_3CHOO$ + water vapour in their experimental setup.
For larger CI ($\geq C_9$) the acid / ester is considered to be fully stabilised, if two esters can be formed they are
considered equally likely. This is recognised as an area where detailed experimental studies are required, to
establish the sensitivity of acid / ester stabilisation to CI size, as well as identifying decomposition products for a
range of CI sizes / structures, and whether these are different for chemically activated / thermalized dioxiranes, as
predicted (Anglada et al., 1998; Nguyen et al., 2009a, 2009b). For $CH_2OO$ decomposition, the protocol assigns
the products equally to two decomposition pathways: $H_2+CO_2$ and $H_2O+CO$; as discussed above, no OH is formed
directly.

For 1,4-and 1,6 allyl H-migration in unsaturated CI (Figure 7), formation of the alkyl radicals from each

of the delocalized radical sites formed after OH elimination is assumed to be equally likely. The product yields
given in Table 2 are for mechanisms that do not explicitly preserve stereo-specificity. For systems that track
stereo-specific substitution on double bonds, H-migration is only possible from the *Z*-substituent, and the number
of products is reduced accordingly, with a concomitant adjustment of the product yields.



For 1,5 ring closure (Figure 8), formation of the epoxide or the dicarbonyl are considered equally likely.
The dicarbonyl undergoes further decomposition to yield two $RO_2$ following Barber et al. (2018). Unimolecular
reaction rates for stabilised unsaturated CI are taken from the Vereecken et al. (2017) SAR. Clearly there remains
much uncertainty on the proposed kinetics, and systematic experimental work on SCI yields, and final product
studies of ozonolysis of conjugated alkenes is required to improve the proposed protocol.
**Table 2. Decomposition pathways and products for CI in the protocol**

| Decomposition Pathway | CI Structure | Products |
|---|---|---|
| *1,4 H-shift (VHP)* | | $+ OH$ |
| *1,3 ring closure (hot acid / ester) CI < C₉* | | $R_1R_2 + CO_2$ (40 %)<br>$R_1OR_2 + CO$ (20 %)<br>$R_1 + R_2 + CO_2$ (40 %) |
| *1,3 ring closure (hot acid / ester) CI ≥ C₉* | | $R_1CO\text{-}O\text{-}R_2$ (50%)<br>$R_1\text{-}O\text{-}COR_2$ (50%) |
| *CH₂OO* | | $H_2 + CO_2$ (50 %)<br>$H_2O + CO$ (50 %) |
| *1,5-ring closure* | R₄ = H or a C atom not bearing an abstractable H | |
| *1,5-ring closure (R₃ = H)* | R₄ = H or a C atom not bearing an abstractable H | (50%)<br><br>$R_1C(O)O_2 + CHOC(O_2)R_2$ (50%) |
| *1,6 allyl H-shift* | | (50%)<br><br>(50%) |
| *1,4 allyl H-shift* | | (50%)<br><br>(50%) |

## 5    Bimolecular Reactions of SCI

Based on the unimolecular pathways described in Section 5, many SCI have lifetimes against unimolecular
reaction on the order of $10^{-3} - 10^{-1}$ s. These lifetimes are long enough to allow them to participate in bimolecular
reactions with trace gases in the atmosphere under typical boundary layer conditions, where Vereecken et al.
(2017) estimated that just under half of the CI in the atmosphere react with a co-reactant rather than
unimolecularly. The co-reactants for which fast reactions, of potential tropospheric importance, have been
demonstrated are $H_2O$, $(H_2O)_2$, $SO_2$, $NO_2$, and organic and inorganic acids (Reactions 6 – 11).





| | | |
|---|---|---|
| 684 | $R_1R_2COO + H_2O \rightarrow x.R_1R_2C(OH)(OOH) + y.R_1C(O)R_2 + z.R_1COOH$ [†] | (R6a) |
| 685 | $R_1R_2COO + (H_2O)_2 \rightarrow x.R_1R_2C(OH)(OOH) + y.R_1C(O)R_2 + z.R_1COOH$ [†] $+ H_2O$ | (R6b) |
| 686 | [†] only available if $R_2 = H$ | |
| 687 | $R_1R_2COO + R_3COOH \rightarrow R_3C(O)OCR_1R_2(OOH)$ | (R7) |
| 688 | $R_1R_2COO + SO_2 \rightarrow R_1R_2CO + SO_3$ | (R8) |
| 689 | $R_1R_2COO + NO_2 \rightarrow R_1R_2C(O_2)ONO$ | (R9) |
| 690 | $R_1R_2COO + HCl \rightarrow ClR_1R_2OOH$ | (R10) |
| 691 | $R_1R_2COO + HNO_3 \rightarrow NO_3R_1R_2OOH$ | (R11) |

Reactions with other trace gases have been investigated both experimentally and theoretically, but these are not
included in the protocol at this time as they are not considered to be important under tropospheric conditions.
Theoretical and experimental work has also shown that more complex bimolecular and unimolecular pathways
may operate forming heterocyclic molecules like cyclic peroxides and secondary ozonides (Chuong et al., 2004;
Long et al., 2019). Again though, these reactions appear to be of negligible importance in the gas phase for SCI
with carbon numbers up to $C_{10}$ (monoterpenes) and are not considered in this protocol. While only reactions
relevant to the atmosphere are included in the protocol; reactions that are not expected to be relevant in the
atmosphere are still maintained in the database since they may be useful to interpret results of chamber simulations
or other laboratory experiments (e.g. self-reaction or reaction with parent alkenes).
$CH_2OO$ and (*E*)-RCHOO react rapidly with $H_2O$ (Reaction R6a) (Welz et al., 2012; Taatjes et al., 2013;
Stone et al., 2014) and with the water dimer, $(H_2O)_2$, (Reaction R6b) (Berndt et al., 2014a; Chao et al., 2015;
Lewis et al., 2015; Lin et al., 2016), such that removal by water vapour is their predominant fate in the atmosphere.
However, (*Z*)-RCHOO react slowly with $H_2O$ (Taatjes et al., 2013; Sheps et al., 2014; Huang et al., 2015)
increasing the importance of bimolecular reactions with other atmospheric trace species such as acids and $SO_2$
(Newland et al., 2018). The reaction of SCI with organic acids (Reaction R7) is also likely to be an important
reaction in the atmosphere (Welz et al., 2014). The experimentally determined reaction rates for SCI + HCOOH
and $CH_3COOH$ are $1 - 5 \times 10^{-10}$ cm$^3$ s$^{-1}$ (Welz et al., 2014; Sipilä et al., 2014; Chung et al., 2019), close to the
collisional limit. Other potentially important reactions in the atmosphere include those with $SO_2$ (Reaction R8),
$NO_2$ (Reaction R9), and inorganic acids (Reactions R10-R11). The rates of SCI+$SO_2$ reaction have been the
subject of several studies for the three smallest SCI, with good agreement between experiments. Larger SCI appear
to have similar reaction rates with $SO_2$ (Ahrens et al., 2014).
The products of many of the bimolecular reactions of SCI are still uncertain. This is the case for the most
important bimolecular reactions in the atmosphere, those with $H_2O$ and $(H_2O)_2$. A recent experimental study
(Sheps et al., 2017) of the reaction of $CH_2OO$ with the $(H_2O)_2$, generating $CH_2OO$ from the photolysis of
diiodomethane, determined yields of: hydroxymethylhydroperoxide (HMHP) (55 %), HCHO (40 %), and
HCOOH (5 %). However, ozonolysis experiments (e.g. Nguyen et al., 2016) have generally found HMHP and
HCOOH to be the main detected products, with negligible yields of HCHO. Based on results from isoprene
ozonolysis chamber experiments, Nguyen et al. (2016) proposed yields from the $CH_2OO + H_2O$ reaction of:
HMHP (73 %), HCOOH (21 %), HCHO (6 %); and from the $(H_2O)_2$ reaction of: HMHP (40 %), HCOOH (54 %),
HCHO (6 %). These low HCHO yields are in agreement with earlier work (Hasson et al., 2001b) that determined
an HCHO yield of $6 - 9$ %.



The products of SCI reaction with organic acids appear to be mainly hydroperoxide esters (Reaction R7).
Hydroperoxy methyl formate (HPMF) has been detected as an intermediate in the $CH_2OO+HCOOH$ reaction (e.g.
Neeb et al., 1995; Wolff et al., 1997; Hasson et al., 2001a; Chung et al., 2019), hydroperoxy methyl acetate in the
$CH_2OO+CH_3COOH$ reaction (Neeb et al., 1996), and hydroperoxy ethyl formate in the $CH_3CHOO+HCOOH$
reaction (Neeb et al., 1995; 1996; Cabezas and Endo, 2020). Theoretical calculations have predicted the formation
of > 90 % HPMF for the reaction of $CH_2OO$ with HCOOH (Vereecken, 2017), and that the production of stabilised
hydroperoxide esters will be even higher for larger SCI. The reaction with $SO_2$ has been shown to form $SO_3$ with
close to unit yield (Reaction R8) (Kuwata et al., 2015). For $NO_2$, while early experimental work (Ouyang et al.,
2013) suggested SCI would oxidise $NO_2$ to $NO_3$, more recent experimental (Caravan et al., 2017) and theoretical
(Vereecken and Nguyen, 2017) work has suggested the formation of a nitroalkylperoxy radical ($R_1R_2C(O_2)NO_2$).
Subsequent reaction and formation of the alkoxy radical would be expected to yield a carbonyl and $NO_2$. The
main products of reaction of SCI with the inorganic acid HCl have been predicted to be chlorohydroperoxides
(Reaction R10) (Foreman et al., 2016; Vereecken, 2017), with these products observed experimentally for
$CH_2OO+HCl$ (Cabezas and Endo, 2017) and $CH_3CHOO+HCl$ (Cabezas and Endo, 2018). The main product of
reaction with $HNO_3$ has been predicted to be hydroperoxynitrates (Reaction R11) (Foreman et al., 2016;
Raghunath et al., 2017; Vereecken, 2017). Raghunath et al. (2017) further predicted decomposition of a fraction
of the chemically activated hydroperoxynitrates to $CH_2(O)NO_3$ + OH. This reaction has not yet been studied
experimentally to the authors' knowledge.

### 5.1    Protocol Rules for SCI Bimolecular Reactions

Bimolecular reaction rate coefficients for SCI are included for reaction with water vapour monomers and dimers,
$SO_2$, $NO_2$, carboxylic acids and inorganic acids (HCl, $HNO_3$) (Table 3). For the water vapour reactions, the rate
coefficients are taken from the SAR of Vereecken et al. (2017), which provides values for 98 explicit structures.
For bimolecular reactions of SCI with the other trace gases, four classes of SCI are considered: $CH_2OO$, *Z/E*-
RCHOO and $R_1R_2COO$ (where R represents alkyl groups), based on the limited experimental data available. The
rates are taken from IUPAC recommendations (Cox et al., 2020) where available, otherwise from sources as stated
in Table 3. Where the structure does not fit into the defined classes, the $CH_2OO$ rate constant is attributed by
default. Reaction products are as given in Reactions R6 – R11. In light of the current uncertainties of the product
distribution of the reactions of SCI with water, here we assume the same products for the monomer and dimer
reactions. We propose yields based on the direct study of Sheps et al. (2017) of α-hydroxyhydroperoxide (55 %),
carbonyl (40 %) and acid (5 %), with the exception of $R_1R_2COO$, which cannot form the acid, for which we
increase the α-hydroxyhydroperoxide to 60 %. These recommendations will be subject to change upon further
experimental information becoming available.






Table 3. Bimolecular reaction rates with RCOOH, SO₂, NO₂ and inorganic acids applied to the four SCI structures.
Rates are IUPAC recommendations (Cox et al., 2020) unless otherwise stated. Bimolecular reaction rates with water
are taken from Vereecken et al. (2017), see main text.

| | Bimolecular reaction rates ($10^{11}$ molecules cm$^{-3}$ s$^{-1}$) | | | | |
|---|---|---|---|---|---|
| | RC(O)OH | SO₂ | NO₂ | HCl[a] | HNO₃[a] |
| CH₂OO | 12 | 3.7 | 0.3 | 4.6 | 54 |
| (E)-RCHOO[b] | 38[c] | 14 | 0.2 | 4.6 | 54 |
| (Z)-RCHOO[b] | 21[c] | 2.6 | 0.2 | 4.6 | 54 |
| R₁R₂COO | 31 | 16 | 0.2 | 4.6 | 54 |

[a] All values for CH₂OO reaction from Foreman et al. (2016); [b] IUPAC recommended values for (E) and (Z)-CH₃CHOO; [c]
Mean of IUPAC recommended values for reaction with HCOOH and CH₃COOH.

## 765  6  Example of protocol application

An example is described below for the unsaturated ketone, 6-methyl-5-hepten-2-one, and illustrated in Figures 9
and 10. Further examples for α-pinene, *cis*-2-pentene, 2-methyl-1-pentene and 2-methyl-1,3-butadiene (isoprene)
are given in the Supplementary (Section S3). The initial rate of reaction with ozone is defined by the protocol in
the companion paper (Jenkin et al., 2020). The branching ratio for formation of the disubstituted CI* is calculated
to be 0.72 using the group additivity values in Table 2 and Eq. (1).

$$Y_{CI1} = \frac{(0.218+0.218)-(0+0)+1}{2} = 0.72 = 1 - Y_{CI2} \qquad (E3)$$

The *syn* and *anti*-conformers of the two large CI* are formed with equal yield (0.14).
Stabilisation of each CI* is computed using Eq. (2):

**(CH₃)₂COO:** $\qquad S = 1 - \left[ \left( \frac{5}{12} \right) \times 1.242 \times \left( \frac{5}{5+(5-5)} \right) \right] = 0.48 \qquad (E4)$
**(Z)-CH₃C(O)(CH₂)₂CHOO:** $\qquad S = 1 - \left[ \left( \frac{8}{12} \right) \times 1.242 \times \left( \frac{5}{8+(5-5)} \right) \right] = 0.48 \qquad (E5)$
**(E)-CH₃C(O)(CH₂)₂CHOO:** $\qquad S = 1 - \left[ \left( \frac{8}{12} \right) \times 0.95 \times \left( \frac{5}{8+(5-3)} \right) \right] = 0.68 \qquad (E6)$

The remaining (CH₃)₂COO* undergoes unimolecular decomposition via the vinylhydroperoxide (VHP) pathway
to yield the acetonyl peroxy radical (CH₃C(O)CH₂OO) and OH. The remaining (Z)-CH₃C(O)(CH₂)₂CHOO
decomposes via the VHP pathway to yield CH₃C(O)CH₂CH(O₂)CHO + OH, while (E)-CH₃C(O)(CH₂)₂CHOO
decomposes via 1,3 ring closure and yields CH₃C(O)CH₂CH₃ + CO₂ (40%), CH₃C(O)CH₂CH₂OH + CO (20%),
CH₃C(O)CH₂CH₂ + H + CO₂ (40%). Each stabilised CI can decompose via the same pathways as its respective
CI*, with temperature dependent rates determined from Vereecken et al. (2017). At 298K these are 478 s⁻¹, 205 s⁻
¹ and 74 s⁻¹ for (CH₃)₂COO, (Z)-CH₃C(O)(CH₂)₂CHOO and (E)-CH₃C(O)(CH₂)₂CHOO respectively.
Alternatively, they can undergo bimolecular reaction. Reaction rates with H₂O and (H₂O)₂ are calculated using
monomer and dimer reaction rates from Vereecken et al. (2017). Reaction rates with other trace gases are taken
from Table 3 for the relevant CI structure. Figure 10 shows calculated pseudo first order reaction rates for reaction
with SO₂ and RCOOH assuming atmospheric mixing ratios of [SO₂] = 5 ppbv and [RCOOH] = 5 ppbv.





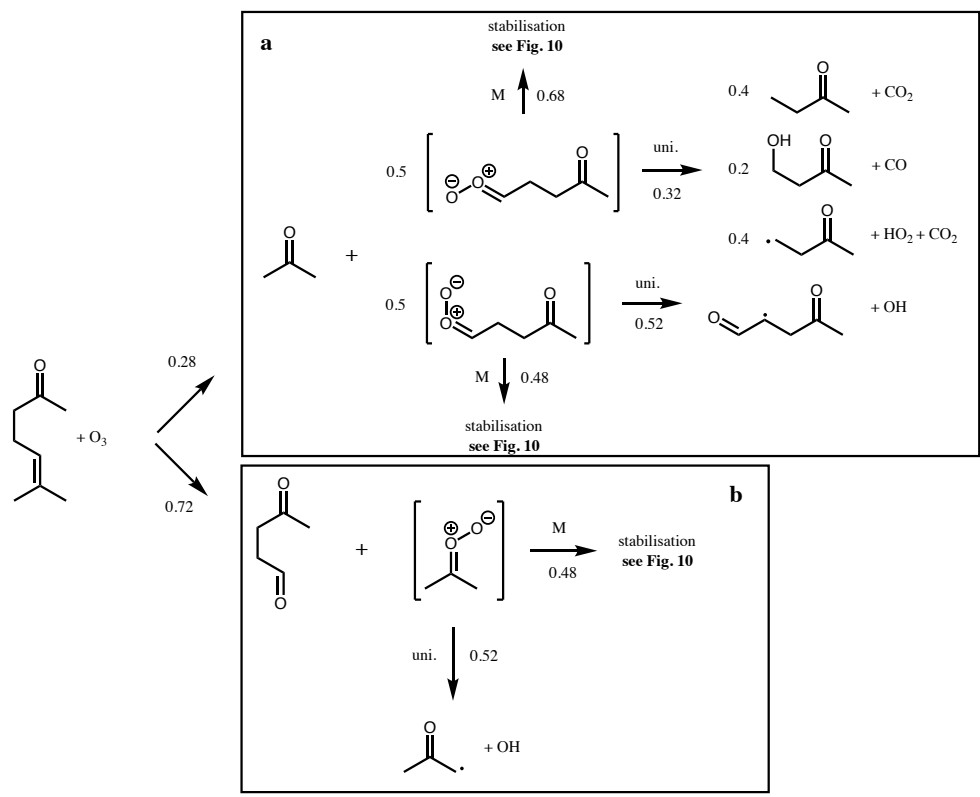

**Figure 9. Branching ratios and products of the CI decomposition produced following ozonolysis of 6-methyl-5-hepten-**
**2-one**






**Figure 10. Bimolecular rate coefficients (see Table 3) and products of the SCI produced following ozonolysis of 6-**
**methyl-5-hepten-2-one at 298 K. Pseudo first order loss rates ($k'$) and products are shown for decomposition and**
**reaction with water vapour, and for other pathways that contribute more than 1% of the total loss assuming [SO$_2$] = 5**
**ppbv, [RCOOH] = 5 ppbv, [H$_2$O] = 5×10$^{17}$ cm$^{-3}$, [(H$_2$O)$_2$] = 5×10$^{14}$ cm$^{-3}$, [NO$_2$] = 1 ppbv, [HCl] = 100 pptv, [HNO$_3$] =**
**100 pptv.**
**[a] 7.64×10$^{-60}$×T$^{23.59}$e$^{(2367/T)}$**
**[b] Sum of first order loss rates to water monomer (7.54×10$^{-18}$ * [H$_2$O]) and dimer (1.82×10$^{-14}$ * [(H$_2$O)$_2$])**
**[c] 2.41×10$^{-62}$×T$^{24.33}$e$^{(2571/T)}$**
**[d] Sum of first order loss rates to water monomer (1.51×10$^{-18}$ * [H$_2$O]) and dimer (4.31×10$^{-15}$ * [(H$_2$O)$_2$])**
**[e] 1.57×10$^{10}$×T$^{1.03}$e$^{(-7464/T)}$**
**[f] Sum of first order loss rates to water monomer (1.58×10$^{-14}$ * [H$_2$O]) and dimer (1.75×10$^{-11}$ * [(H$_2$O)$_2$])**
**7    Protocol Evaluation**
**7.1    Experimental databases and assessment approach**
A database of experimentally determined carbonyl yields, OH yields and SCI yields has been assembled in order
to evaluate the new protocol (Supplement – Spreadsheets S1-S3). Experimental conditions are also recorded in
the database to enable some assessment of the validity of the assumptions inherent in the experimental setup.

The Root Mean Squared Error (RMSE) and the Mean Bias Error (MBE) were examined to assess the

reliability of the protocol. The RMSE and MBE are here defined as:

$$RMSE = \sqrt{\frac{1}{n}\sum_{i=1}^{n}\left(Y_{protocol} - Y_{database}\right)^2} \qquad (E7)$$





$$MBE = \frac{1}{n}\sum_{i=1}^{n}\left(Y_{protocol} - Y_{database}\right) \qquad (E8)$$
where $n$ is the number of species in the dataset. The databases were split in to subsets to identify possible bias
within a structural category of species (e.g. exocyclic vs endocyclic monoalkenes). The various subsets examined
and their corresponding number of species are summarized in Table 4. Three databases were used to perform the
protocol assessment: carbonyl yields (Spreadsheet S1), SCI yield (S2), and OH yield (S3). The RMSE and MBE
computed for the full databases and the various subsets are reported in Table 4. The scatter plots of protocol yields
vs database yields, by species category, are given in Figure 11 .

**Table 4. Number of species (n) in the database used to compute the mean bias error (MBE) and the root mean square**
**error (RMSE) for the OH yields, SCI yields and "longest" carbonyl yield.**

| | all species | acyclic monoalkene | endocyclic monoalkene | exocyclic monoalkene | poly alkene | aromatic alkene | oxygenated alkene |
|---|---|---|---|---|---|---|---|
| *OH yields* | | | | | | | |
| *n* | 46 | 18 | 8 | 3 | 10 | 3 | 4 |
| **MBE** | 0.02 | -0.01 | -0.03 | -0.01 | 0.13 | 0.03 | 0.01 |
| **RMSE** | 0.13 | 0.06 | 0.09 | 0.12 | 0.23 | 0.09 | 0.16 |
| *SCI yields* | | | | | | | |
| *n* | 22 | 11 | 5 | 2 | 3 | 1 | 0 |
| **MBE** | 0.05 | 0.02 | -0.01 | 0.22 | 0.01 | 0.45 | - |
| **RMSE** | 0.12 | 0.04 | 0.04 | 0.22 | 0.06 | 0.45 | - |
| *Yields of the longest carbonyl* | | | | | | | |
| *n* | 73 | 35 | NA[‡] | 5 | 5 | 1 | 27 |
| **MBE** | -0.01 | -0.02 | - | -0.04 | 0.03 | -0.02 | 0.00 |
| **RMSE** | 0.14 | 0.11 | - | 0.12 | 0.07 | 0.02 | 0.18 |

[‡] Endocyclic alkenes do not produce a stable primary carbonyl as all possible molecules formed from the POZ fragmentation
contain a carbonyl oxide moiety.

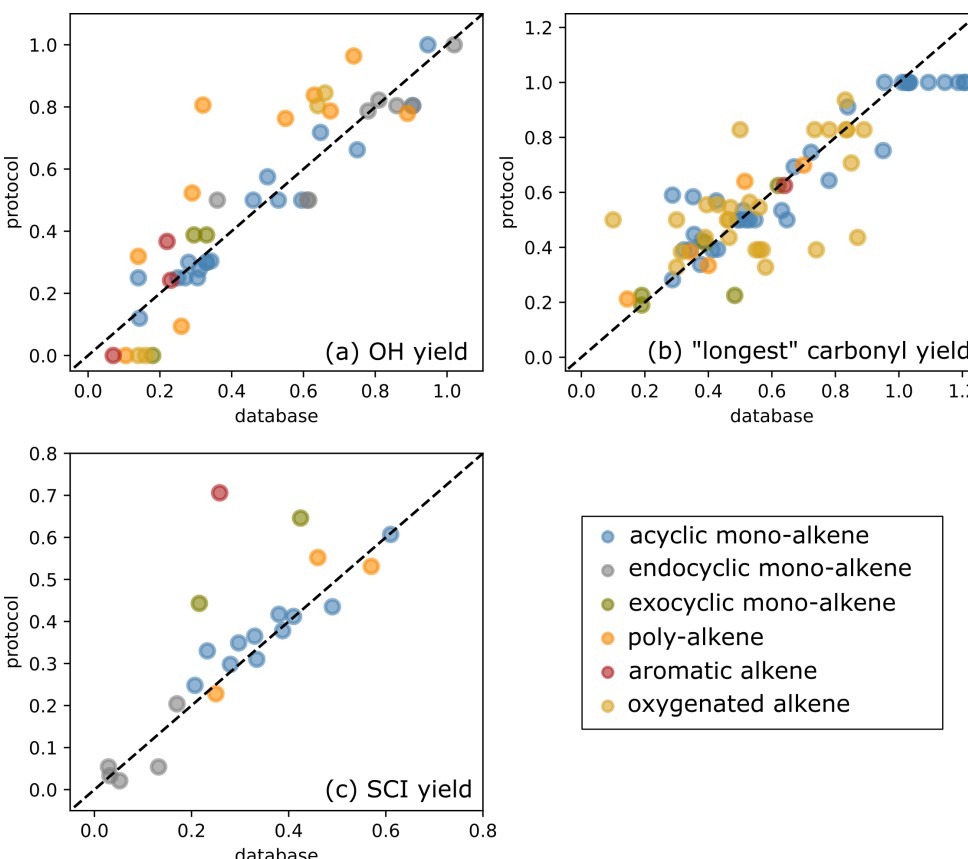

**Figure 11. Scatter plot of protocol yields vs database yields: (a) OH yields, (b) yields for the "longest" carbonyl and (c) SCI yields.**

### 7.2    Primary Carbonyl Yields

The primary carbonyl yields from alkene ozonolysis are calculated in the protocol by assigning $F$ values to different functional groups adjacent to the C=C bond that determine the relative fragmentation pattern of the POZ (Section 2). The calculated primary carbonyl yields can be compared to the measurements in the experimental database. For some functional groups however, the number of data available is sparse and the carbonyl yields have been directly used to determine the $F$ value. The carbonyl yields dataset should therefore rather be viewed as a training dataset than a validation dataset in this protocol assessment. Figure 11b shows the scatter plots for the calculated yields of the larger primary carbonyl (i.e. greater number of non-H atoms) formed in POZ decomposition, compared to the experimentally reported values for each alkene in the database. No substantial bias is identified in the computed carbonyl yields (MBE=-0.01). For non-oxygenated alkenes, the fit is reasonably good and the RMSE does not exceed 0.12 for the various hydrocarbon classes reported in Table 4. The major outlier is the yield of 3-methyl-2-pentanone from 3,4-diethyl-2-hexene ozonolysis. This is based on one measurement (Grosjean and Grosjean, 1997). It was noted in Jenkin et al. (2020) that the ozonolysis reaction rate reported by Grosjean and Grosjean (1997) for this compound was a significant outlier from predicted trends, and so it seems possible that this compound was incorrectly identified in the original work. For symmetrical alkenes,



the calculated primary carbonyl yield is unity, whereas measured yields tend to cluster slightly above one. This is
likely due to a small amount of secondary formation of the carbonyls from bimolecular reactions of SCI. The
poorest fitted class is oxygenated alkenes (RMSE=0.18). This is likely due to a combination of factors. Firstly,
the majority of these compounds have only one measurement. Secondly, measurements of multi-oxygenated
VOCs are known to be more challenging than e.g. simple carbonyls. Thirdly, there is more likely to be additional
chemical factors which are not yet understood in the ozonolysis of these more complex molecules influencing the
POZ fragmentation. Two of the most significant outliers in the oxygenated alkenes are acrylic and methacrylic
acid. As described in Section 2.2.3 it is difficult to reconcile the two available data points.
### 7.3    SCI yields
The yield of stabilised Criegee intermediates from an alkene-ozone reaction depends on the alkene structure (i.e.
the POZ fragmentation pattern, Section 2), the dominant unimolecular decomposition route of the CI* (Section
3.2), and the size of the CI (Section 3.2). The yields calculated by the protocol are independent of the
measurements in the database. SCI yields can therefore be considered as a validation dataset to evaluate the
reliability of the protocol. Total SCI yields have been measured for a number of alkenes, although the dataset is
still relatively small. It should also be noted that many experimentally determined SCI yields have a large
uncertainty associated with them, particularly earlier experiments where analysis techniques were less developed
and the chemical models lacking. Figure 11c shows the scatter plot of the total SCI yields calculated by the
protocol vs experimental data. The data consists predominantly of acyclic monoalkenes, for which there is a good
agreement between the measurements and the calculated values (RMSE=0.04). Figure 11c shows three major
outliers for which the protocol over-predicts the measured SCI yield. These species are methylene cyclohexane
and β-pinene (which constitute the subset of the exocyclic alkenes (RMSE=0.22)) and styrene, the only
representative of the aromatic alkenes class in this dataset (RMSE=0.45). The methylene cyclohexane and styrene
values are both based on one measurement (Hatakeyama et al., 1984), and the β-pinene value is based on two
measurements (Hatakeyama et al., 1984; Newland et al., 2018) which are in poor agreement, giving values of 0.25
and 0.60 respectively. This clearly warrants revisiting experimentally, particularly with respect to the
atmospherically important monoterpene β-pinene. Finally, the overall protocol SCI yields appear to be biased
slightly high (+ 5%), which is mainly explained by the overestimation described above for the exocyclic and
aromatic alkenes.
### 7.4    OH yields
The reaction of alkenes with ozone yields OH through both primary (i.e. decomposition of CI via a
vinylhydoperoxide) and secondary (i.e. peroxy radical chemistry) processes. The primary process can also be
split: the decomposition of chemically activated CI*, which under atmospheric conditions (and e.g. chamber
laboratory experiment conditions) is assumed to happen at rates such that there is no competition with bimolecular
reaction; and the decomposition of stabilised CI, which occurs in competition with bimolecular reactions so that
the OH yield depends on the unimolecular rate relative to the concentrations of possible co-reactants. The primary
OH yield thus depends on the POZ fragmentation pattern (Section 2) and the decomposition pathways of the CI
(Section 3.2).



Many studies have measured the OH yield for specific alkene-ozone reactions. As for the SCI yields
above, the OH yield database can be viewed as a validation dataset to assess the reliability of the protocol since
OH yields are not prescribed explicitly, but are a product of the protocol rules for POZ fragmentation and CI
decomposition pathways. For the comparison, protocol yields are computed assuming that all SCI produced
undergo unimolecular decomposition (i.e. bimolecular reactions of SCI are ignored). Although many experiments
will have been designed in such a way as to try to prevent bimolecular reactions, in reality a small fraction of the
SCI will react bimolecularly, not producing OH, so the computed OH yield might be considered an upper limit.
On the other hand, in many of the experiments there will likely be some contribution to the measured OH yield
from peroxy radical chemistry (e.g. $HO_2 + O_3$), making the reported experimental yield an upper limit. No attempt
is made here to determine the relative contribution from primary or secondary processes in the reported
measurements, which is dependent on both experimental setup and the particular alkene being studied, or to
correct for possible bimolecular reactions. Therefore, a comparison between experimental and protocol OH yields
clearly carries significant uncertainties.

With this in mind, the agreement between computed OH yields and the experimental values is very good
(Figure 11a). No substantial bias is observed on the complete dataset (MBE=0.02). It is difficult to comment on
some classes as they contain only one or two compounds (see Table 4). The protocol appears especially reliable
for estimating the OH yields for monoalkenes (RMSE=0.06) and endocyclic alkenes (RMSE=0.09). The class for
which the protocol does worst is polyalkenes (RMSE=0.23), with a systematic over-prediction at higher OH yields
(MBE=0.13). There are five compounds for which the protocol calculates an OH yield of zero (styrene, 1,3-
butadiene, methyl vinyl ketone, methacrolein, and camphene). The measured OH yields of these compounds are
all below 0.2 and the measured OH could be a result of peroxy radical chemistry.
**8    Conclusions**
This manuscript provides a protocol by which the central features of alkene ozonolysis chemistry can be included
in an explicit automatic chemical mechanism generator. It also serves to highlight the many gaps that remain in
our knowledge of this complex, atmospherically important, process. This will hopefully help direct both
experimental and theoretical research towards improving understanding in these areas. Some of the major areas
of uncertainty identified in this work include:
(i)     The impact of oxygenated substituents on POZ fragmentation
(ii)    The impact of alkene structure on $(E)/(Z)$-CI conformer yields
(iii)   Products of the hot acid / ester channel and trends in the stabilisation of the hot acid / ester with size
(iv)   Further details of the mechanisms and products of non-Criegee ozonolysis chemistry, e.g. step-wise
decomposition of the POZ via a carbonyl hydroperoxide
(v)     Product distributions of some of the major atmospheric SCI bimolecular reactions – e.g. the reaction
of $(Z)$-RCHOO / $CH_2OO$ with $H_2O$ / $(H_2O)_2$
(vi)   Experimental evidence of the products of conjugated alkene ozonolysis
(vii)  Data on OH and SCI yields from alkenes with (multiple) functional groups
The reliability of the protocol designed in this work was assessed using experimental values for the OH, SCI and
primary carbonyl yields. For these three datasets, the Mean Bias Error (MBE) for the protocol based yields is



below 0.05, with no substantial bias identified. The protocol currently provides a fairly reliable estimate of the
OH, SCI and primary carbonyl yields with Root Mean Squared Errors (RMSE) of 0.12, 0.13 and 0.15,
respectively. However, the number of data available for some classes of compounds remains limited, such as
oxygenated, exocyclic and poly-alkenes. The errors in the yields calculated for these species are also the most
substantial, and additional experimental data for these categories of compound would be highly valuable to
improve the protocol and its assessment.
**Acknowledgements**
This work was performed as part of the MAGNIFY project, with funding from the French National Research Agency (ANR)
under project ANR-14-CE01-0010, and the UK Natural Environment Research Council (NERC) via grant NE/M013448/1. It
was also partially funded by the European Commission (EUROCHAMP-2020 (grant no. 730997)) and the UK National Centre
for Atmospheric Sciences (NCAS) Air Pollution Theme.
**Special issue statement**
This article is part of the special issue "Simulation chambers as tools in atmospheric research (AMT/ACP/GMD inter-journal
SI)". It is not associated with a conference.

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
