# Peer review of "Estimation of mechanistic parameters in the gas-phase reactions of 1"

_Atmospheric Chemistry and Physics, 2021_

## Author Comment (AC1)

**Authors' responses to referee comments on: Newland et al., Atmos. Chem. Phys. Discuss., https://doi.org/10.5194/acp-2021-1031.**

**Author Amendments**

Note that the discussion on in Section 4.4 Unimolecular reactions of unsaturated CI on dioxole formation pathways from *syn*-MACRO and *anti*-MVKO now quotes the separate calculated rates for each species as given in Vereecken et al. 2017:

*"The calculated unimolecular decay rates for the dioxole forming pathways from syn-MACRO and anti-MVKO are fast; Vereecken et al. (2017, Table 25 in supplementary material) reported rates of 2500 and 7700 s$^{-1}$, respectively, with increasing substitution on the vinyl group accelerating the reaction further, while Barber et al. (2018) reported a somewhat slower rate for anti-MVKO of 2140 s$^{-1}$."*

Additional typographical errors found by the authors have been corrected in the revised manuscript.

Anonymous Referee #1

**https://doi.org/10.5194/acp-2021-1031-RC1**

We are very grateful to the reviewer for their supportive comments on this work, and for their helpful suggestions for mechanistic clarifications. Responses to the comments are provided below (the original comments are shown in black font with responses in red).

**General comments**

This manuscript continues the important work of enabling an automated mechanism construction for alkene ozonolysis. The authors focus on predicting among the most important atmospheric impacts of ozonolysis, namely primary carbonyl, OH, and stabilized Criegee intermediate (SCI) yields. The work provides a comprehensive and critical review of the many facets of the ozonolysis of the major classes of alkenes and helpfully highlights important knowledge gaps. The presentation of chemical rationales for trends in yields is largely illuminating. The authors do a good job of clearly presenting current shortcomings of the protocol being presented here, providing other investigators a good sense of the important outstanding questions. In sum, this manuscript makes an important contribution to mechanistic atmospheric chemistry.

**Specific comments**

Section 2.1 implies that the tertiary C in a tert-butyl group bears a partial positive charge, significantly lowering the yield of the tert-butyl substituted CI. The authors should explain more how the tertiary C is partially positive, given all of the hyperconjugation of the C-H bonds.

Fliszár & Renard 1970 state that "the zwitterion which is formed preferentially is the one whose environment is better able to stabilize the positive charge by increasing the electron density in the vicinity of the potential zwitterionic carbo-cation (in the transition state) via inductive and mesomeric effects". Here, it is not the tertiary C in the *tert*-butyl group (β-carbon) itself that is partially positive, but the α-carbon in the POZ transition state that the group is attached to, which leads to a zwitterion upon decomposition. The difference in the relative hyperconjugative and inductive effects of the substituents groups on either side of the double bond determines the relative proportions of the each Criegee zwitterion formed. For tabulated quantitative data on such effects, we refer to the group additivity factors given in Table 16 in the SI of Vereecken et al. 2017, showing that these substituents all lead to CI more stable

than with H-substituents, but that for alkyl, vinyl, and γ-unsaturated substituents the stability decreases with increased branching on the β-carbon, i.e. H-atoms on the β-carbon are more effective in stabilizing the positive charge than branching alkyl groups on the β-carbon.

We have added the following to the end of Section 2.1 to clarify these points further:

*"Finally, Vereecken et al. (2017, Table 16 in the supplementary material) analysed the stability of CI in terms of group additivity factors, showing that alkyl-substituted CI are more stable than H-substituted CI, but where the stability of the CI is inversely proportional to the branching on the β-carbon atom."*

In Section 2.3, it would be worthwhile to rationalize briefly the destabilizing effects of a vinyl group compared to a H, methyl, or isopropyl group.

We have rephrased this section, as the conjugation between a vinyl group and the carbonyl oxide π-system (C=C-C=O+O-) leads to more stable CI compared to a H-atom substituent (again, see Table 16 in the SI of Vereecken et al. 2017), while in the POZ decomposition TS the vinyl group is less favourable than an H-atom as this conjugative effect is not available yet (no double bond yet in what will become the carbonyl oxide moiety).

As to why a vinyl group is less favourable than an H-atom or alkyl group in the POZ transition state is due to all the specific substituent-specific interactions, i.e. a trend analysis purely based on stabilization of the positive charge becomes untenable between non-homologous substituent groups (H-atom, alkyl group, vinyl group, oxygenated group,…). Vereecken et al. 2017 explicitly remarks on the strong impact of the interactions between H-atoms on a β-carbon and the syn-carbonyl oxide outer O-atom; as a vinyl group has very different geometric parameters than an alkyl group, any alkyl-based trend is not expected to directly transfer to vinyl substituents, or between alkyl, H-atoms, and vinyl groups.

We have added the following to the end of Section 2.3 to clarify these points further:

*"Note that, once the CI is formed, the vinyl group can conjugate with the carbonyl oxide π-system, leading to additional stabilization such that vinyl-CI are more stable than H-substituted CI (Vereecken et al. 2017); this is however a product-specific effect that is not available yet in the POZ decomposition."*

In Section 2.8.3, contrary to what the authors assert, I think that there is some reason to question the generality of the OH formation from the carbonyl-hydroperoxide pathway. The reason is that radical formation requires the hydroperoxide to be significantly chemically activated, and larger hydroperoxides will be more prone to collisional stabilization.

In section 2.8.3, we state that "Currently this channel (the carbonyl-hydroperoxide channel) is only included for the ethene-ozone reaction". This is because it is needed to explain radical formation in the ozonolysis of ethene (for which OH formation via a vinylhydroperoxide intermediate is not available). However, we have also noted here that "There is no reason to assume it (the carbonyl-hydroperoxide channel) will not occur more generally for any alkenes with vinylic H-atom(s), though perhaps with different fates of the intermediate biradical and/or carbonyl hydroperoxide". Therefore, we have extended this statement to include the following:

*"(e.g. larger hydroperoxides could be more prone to collisional stabilisation and yield less prompt OH)".*

When more general data become available, assuming the channel is active for other systems, the protocol will be updated.

**Technical comment**

None.

Anonymous Referee #2

[https://doi.org/10.5194/acp-2021-1031-RC2](https://doi.org/10.5194/acp-2021-1031-RC2)

We thank the reviewer for their broadly positive comments on this work and for their helpful suggestions for text clarifications. We sincerely hope that the work is seen more than just a "review" and that a broad range of atmospheric chemists will use it accordingly in order to understand and build detailed chemical mechanisms for atmospheric ozonolysis (as they have done with the series of other detailed mechanism construction protocols we have published in ACP).

The work also highlights where important gaps in our knowledge remain, and we hope laboratory and theoretical chemists will use it to provide a focus for future studies.

Responses to specific comments are provided below (the original comments are shown in black font with responses in red).

This paper describes a formulism for estimating the yields of Stabilized Criegee Intermediates, primary carbonyls, and OH radicals produced from the ozonolysis of a variety of alkenes. The formulism is designed to assist in the automatic generation of chemical mechanisms for use in, for example, GECKO-A and the Master Chemical Mechanism. The paper is part of a longer series, which describe kinetics and branching ratios for alkanes, alkenes, and aromatics. The paper will, I suspect, be of most use to a small number of people involved in mechanism generation, but should serve as a useful data base on alkene ozonolysis, and could also provide estimation methods for product yields with which to compare new experimental data.

The paper appears more like a review than a critical evaluation of the data available. This is due in part to the limited number of measurements that have been made, and to a lack of precision of older measurements. Consequently, there is sometimes no real basis for the evaluation. However, the authors have done a good job in outlining the limitations of the data set, and in selecting the best values to use. So, I think overall, this is a valuable addition to the literature, and will find usage within the community.

**Some minor comments follow.**

Line 42. Mauldin II should be Mauldin III (although I don't think it's really necessary to put the II or III in the text, just the references should suffice.

Roman numerals distinguishing Mauldin from his Father and Grandfather removed from within the text.

Line 86. "This": Be specific as to what "This" is. The paper? The protocol?

We have replaced "*This*" with "*This protocol*"

Line 121 and Figure 3. I think there's a discrepancy here. The larger carbonyl from 1-butene would be propanal, but the text says the yield is 0.35, while the Figure caption gives 0.64. Please check my logic.

We thanks the reviewer for spotting this inconsistency. We have changed the yield in the Figure caption from 0.64 to 0.35. Figure 3 caption has been further clarified to read:

*"Figure 3. Decreasing order of preference, from left to right, of more substituted CI formation from ozonolysis of example alkyl substituted alkenes. Values are 1 – (mean of measured yields of carbonyls) (Spreadsheet S1). * Mean measured yield of propanal (i.e. 1 – more*

*substituted CI) formation from 1-butene is 0.35, but for all other 1-alkenes the yield of the larger primary carbonyl product ranges from 0.45 – 0.50."*

Line 181. I couldn't find the Wang et al, 2020 paper in the reference list.

We thank the reviewer for spotting this omission. We have also noticed that a reference to the work of Lei et al., 2020 has also been omitted. Both references have been added.

Lei, X., Wang, W., Gao, J., Wang, S. and Wang, W.: Atmospheric Chemistry of Enols: The Formation Mechanisms of Formic and Peroxyformic Acids in Ozonolysis of Vinyl Alcohol, J. Phys. Chem. A, 124, 4271-4279, 2020.

Wang, S., Newland, M.J., Deng, W., Rickard, A.R., Hamilton, J.F., Muñoz, A., Ródenas, M., Vázquez, M.M., Wang, L. and Wang, X.: Aromatic photo-oxidation, a new source of atmospheric acidity, Environmental Science & Technology, 54, 7798-7806, 2020.

Line 195. "The acetate" is strictly an anhydride (but I see why the authors used the terminology).

*We have changed "acetate" to "anhydride".*

Line 358. Please check. Methyl vinyl ketone would be –C(=O)R not –C(=O)H?

*The reviewer is correct, MVK would be -C(=O)CH$_3$. Text changed as appropriate.*

Lines 427-431. Maybe these two sentences could be written with a little more clarity. I think the logic is correct – the larger Criegee takes away more of the energy, but also has a higher probability for stabilization. It is not totally clear from the text which wins. I think line 429 is the key, where it's not clear whether it's talking about nascent CI* or SCI. Please re-read closely.

The original text was indeed unclear. We have rephrased to indicate that larger CI moieties do indeed take a somewhat larger fraction of the POZ energy, but that the energy per degree of freedom decreases, such that overall it leads to less prompt reaction and more SCI formation. We have also added a sentence stating explicitly that a larger carbonyl fragment also reduces the energy in the CI, thus enhancing SCI yields.

*"Larger CI* will tend to be stabilised to a greater extent due to a greater density of states distributing the excess internal energy over a greater number of modes and so reducing the rate of unimolecular decay (Drozd and Donahue, 2011; Stephenson and Lester, 2020). Hence, as the size of the CI increases relative to the carbonyl co-product formed in POZ decomposition, the fraction of the energy taken by the CI from the POZ will increase somewhat (assuming the energy has time to become equally distributed throughout the POZ), but typically the mean excess energy per degree of freedom of the nascent CI population decreases, and hence the fraction of CI* with enough energy to undergo unimolecular decay also decreases (Fenske et al., 2000; Newland et al., 2020). This will lead to greater stabilisation. i.e. higher SCI yields. Similarly, for a given CI size, carbonyl co-products of increasing size will take a larger fraction of the excess energy, leaving the CI* moiety with less energy and thus will also lead to higher SCI yields (Newland et al., 2020)."*

Line 471 is a little ambiguous in the use of respectively (there are three clauses). I think Fvhp applies to (CH3)2COO, and F13c to CH2OO, so can probably just switch these.

We thank the reviewer again for spotting this inconsistency. We have tried to make the sentence structure clearer here:

*"…where $A_{CI}$ is the total number of non-hydrogen atoms in the CI\* and $A_{tot}$ is the total number of non-hydrogen atoms in the POZ. $F_{13RC}$ and $F_{VHP}$ are values determined for CH2OO and (CH3)2COO, based on the SCI yields for their symmetrical parent alkenes ethene and 2,3-dimethylbut-2-ene, respectively"*

Line 689 (reaction R9) Implies the formation of an C–ONO bond, while the text a little later refers to a nitroalkyl peroxy radical, i.e. a C-N bond.

Reaction R9 changed to give the product $R_1R_2C(NO_2)O_2$?

Note, we have also added a new reference with respect to the discussion on the products of $CH_2OO$ + HCl (Reaction R10):

Taatjes, C. A., Caravan, R. L., Winiberg, F. A. F., Zuraski, K., Au, K., Sheps, L., Osborn, D. L., Vereecken, L., and Percival, C. J.: Insertion products in the reaction of carbonyl oxide Criegee intermediates with acids: Chloro(hydroperoxy)methane formation from reaction of CH2OO with HCl and DCl, Mol. Phys., 119, e1975199, https://doi.org/10.1080/00268976.2021.1975199, 2021.

Line 690 (reaction R10) Product missing a carbon atom.

Added.

 Line 691 (reaction R11) Product missing a carbon atom.

Added.

Table 3. Units in top row should be cm^3 molecule^-1 s^-1.

Changed.

Line 836. Not clear how to get 3-methyl-2-pentanone from that precursor anyway.

Here, the product should actually be *4-ethyl-3-hexanone* (taken from Grosjean and Grosjean 1996a).

Note that we have now added references for the following primary literature for the rate constants and product yields for the ozonolysis of 1,1-disubstituted alkenes:

Grosjean, E., and Grosjean, D.:  Carbonyl products of the gas phase reaction of ozone with 1,1-disubstituted alkenes. J. Atmos. Chem., 24, 141–156, https://doi.org/10.1007/BF00162408, 1996a.

Grosjean, E. and Grosjean, D.: Rate constants for the gas phase reaction of ozone with 1,1-disubstituted alkenes, Int. J. Chem. Kinet., 28, 911–918, 1996b.

And the following missing reference:

Grosjean, E. and Grosjean, D.:  Gas phase reaction of alkenes with ozone: Formation yields of primary carbonyls and biradicals. Environ. Sci. Technol., 31(8), 2421-2427, 1997a.